

# VERA-ARAB: unveiling the Arabic tweets credibility by constructing balanced news dataset for veracity analysis

Mohamed A. Mostafa[1] and Ahmad Almogren[2]

[1] Department of Computer Science, College of Computer and Information Sciences, King Saud University, Riyadh, Saudi Arabia
[2] Chair of Cyber Security, Department of Computer Science, College of Computer and Information Sciences, King Saud University, Riyadh, Saudi Arabia

## ABSTRACT

The proliferation of fake news on social media platforms necessitates the development of reliable datasets for effective fake news detection and veracity analysis. In this article, we introduce a veracity dataset of Arabic tweets called "VERA-ARAB", a pioneering large-scale dataset designed to enhance fake news detection in Arabic tweets. VERA-ARAB is a balanced, multi-domain, and multi-dialectal dataset, containing both fake and true news, meticulously verified by fact-checking experts from Misbar. Comprising approximately 20,000 tweets from 13,000 distinct users and covering 884 different claims, the dataset includes detailed information such as news text, user details, and spatiotemporal data, spanning diverse domains like sports and politics. We leveraged the X API to retrieve and structure the dataset, providing a comprehensive data dictionary to describe the raw data and conducting a thorough statistical descriptive analysis. This analysis reveals insightful patterns and distributions, visualized according to data type and nature. We also evaluated the dataset using multiple machine learning classification models, exploring various social and textual features. Our findings indicate promising results, particularly with textual features, underscoring the dataset's potential for enhancing fake news detection. Furthermore, we outline future work aimed at expanding VERA-ARAB to establish it as a benchmark for Arabic tweets in fake news detection. We also discuss other potential applications that could leverage the VERA-ARAB dataset, emphasizing its value and versatility for advancing the field of fake news detection in Arabic social media. Potential applications include user veracity assessment, topic modeling, and named entity recognition, demonstrating the dataset's wide-ranging utility for broader research in information quality management on social media.

# INTRODUCTION

The advent of social media has revolutionized the way information is disseminated, yet it has also paved the way for the rapid spread of fake news, leading to significant social and political consequences worldwide. According to *Allcott & Gentzkow (2017)*, the

Corresponding authors
Mohamed A. Mostafa, momostafa@ksu.edu.sa
Ahmad Almogren, ahalmogren@ksu.edu.sa

widespread dissemination of false information on social media can significantly influence public opinion and behavior, thereby impacting political processes and societal harmony (*Nekmat, 2020*). Social media has emerged as a prominent platform for news consumption, offering cost-free and easily accessible content that can be rapidly disseminated. Consequently, it has become an essential channel for individuals to both share and consume information. A large number of people who use online sources, particularly social media networks, to access news. Consequently, a significant proportion of the population relies on social media as their primary medium for obtaining news (*Perrin, 2015*). The lack of trust in news presents a major challenge for ensuring fairness in information dissemination. This issue is compounded by the difficulty in distinguishing between real and fake news articles, making it increasingly challenging for individuals to receive and consume accurate information (*Torabi Asr & Taboada, 2019*).

Twitter has emerged as one of the most influential platforms for news dissemination, characterized by its ease of sharing and real-time information flow. With over 400 million users worldwide and 330 million monthly active users globally, Twitter serves as a vital medium for both individuals and news organizations to share updates and engage with audiences (*Zote, 2024*).

In the Arabic-speaking countries, Twitter's role is particularly pronounced, with millions of Arabic-speaking users relying on the platform for their daily news consumption. According to a report by University of Oregon (*Radcliffe, Abuhmaid & Mahliaire, 2023*), Saudi Arabia and Egypt are significant markets for Twitter in the Arab region. Saudi Arabia has approximately 12.3 million active Twitter users, while Egypt has around 3.7 million users. Both countries rank within the top 20 globally for Twitter users. The report further indicates that MENA region, particularly Arab countries, experiences high social media usage, with users spending an average of 3.5 h per day on social media platforms, much of which is dedicated to news consumption. Around 79% of Arab youth obtain their news from social media, showing a significant increase from previous years. This substantial user base underscores the importance of focusing on Arabic tweets for fake news detection. Moreover, the linguistic and cultural nuances inherent in Arabic tweets present unique challenges that require specialized approaches for effective fake news detection.

Over the past years, a considerable body of research has focused on the application of machine learning methodologies to develop automated models aimed at detecting fake news on social media platforms. Recent surveys highlight various machine learning techniques (*Mishra, Shukla & Agarwal, 2022*; *Alghamdi, Luo & Lin, 2024*), with some focusing on textual models for news content (*Al-Yahya et al., 2021*; *Himdi et al., 2022*; *Capuano et al., 2023*), while others explore multimodal approaches that incorporate both textual and multimedia features (*Comito, Caroprese & Zumpano, 2023*; *Tufchi, Yadav & Ahmed, 2023*). However, many of these studies primarily focus on English datasets, leaving a gap in research on other languages.

Detecting fake news in Arabic tweets presents several unique challenges due to the linguistic characteristics of the Arabic language. Arabic is a morphologically rich language with complex grammar and a wide variety of dialects, making natural language processing

tasks particularly difficult (*Othman, Al-Hagery & El Hashemi, 2020*). Furthermore, Arabic is considered a low-resource language computational linguistics, with a limited availability of annotated datasets and linguistic tools compared to languages such as English (*Faheem et al., 2024*). The scarcity of robust Arabic datasets specifically designed for fake news detection exacerbates this challenge, as existing resources are either sparse or not publicly accessible (*Alhayan, Himdi & Alharbi, 2024*). Moreover, there is a notable lack of comprehensive studies focusing on fake news detection within Arabic social media content, leaving a significant gap in the research landscape. Addressing these challenges requires the development of specialized resources and methodologies tailored to the linguistic intricacies of Arabic.

The construction of a high-quality dataset is of paramount importance in facilitating the development of reliable machine learning models. A robust dataset serves as the foundation upon which models are trained, validated, and tested. By ensuring the dataset consists accurate, diverse, and representative samples, researchers can enhance the generalizability of their models and mitigate biases. Furthermore, the availability of reliable ground truth labels or annotations within the dataset is crucial for effectively training supervised learning models.

This study aims to enhance fake news detection in Arabic tweets by employing advanced machine learning techniques. By leveraging both traditional feature extraction methods and state-of-the-art word embeddings, the goal is to improve detection accuracy and address the linguistic nuances of Arabic. This study makes several significant contributions to the field of fake news detection in Arabic tweets:

- A benchmark balanced multidomain multidialectal dataset of Arabic tweets, annotated with binary labels indicating fake or true news, has been constructed and consists of approximately 20,000 tweets. The dataset contains information of varacity of Arabic tweets, called VERA-ARAB.
- The dataset construction is meticulously verified by an independent fact-checking third party comprising expert journalists, Misbar, who provide evidence for each annotation to ensure reliability and accuracy.
- Exploratory analysis of the constructed dataset, uncovering insights and patterns from the tweets that can inform further research. By conducting in-depth statistical, spatiotemporal, user profile, and content analysis.
- Extensive learning experiments are proposed that exploit both social features and textual features derived from the tweet content to assess the detection methodology.
- Moreover, an experiment on text embedding using AraVec, a state-of-the-art word embedding model specifically designed for the Arabic language is conducted, to evaluate its effectiveness in enhancing fake news detection.

The remainder of this article is organized as follows: the "Related Works" section presents a comprehensive summary of the relevant literature. The "Data Acquisition Methodology" section explains the process of constructing the dataset and the annotation methodology. "Data Analysis" section, provide an in-depth exploration and insights

extracted from the dataset. The "Fake Arabic News Detection" section addresses the challenge of detecting fake news in Arabic tweets, presenting experimental results and an analysis of various classification algorithms. The "Future Work and Other Applications" section outlines potential future research directions and additional applications that can benefit from the VERA-ARAB dataset. Finally, the article concludes with a summary of the findings in the "Conclusions" section.

## RELATED WORK

The detection of fake news on social media has garnered significant attention from researchers across various fields to its critical impact on public opinion and societal well-being. Early studies primarily focused on English-language datasets, leveraging a range of machine learning techniques to identify false information. Methods such as naive Bayes, support vector machines (SVM), and decision trees have been widely utilized, achieving varying degrees of success depending on the feature extraction methods and the characteristics of the datasets (*Shu et al., 2017*; *Zhou & Zafarani, 2020*).

Despite these advancements, research on fake news detection in non-English languages, particularly Arabic, remains relatively sparse. Arabic presents unique challenges due to its rich morphology, diverse dialects, and script variations. Studies focusing on Arabic fake news detection have started to emerge, utilizing both traditional machine learning and deep learning approaches. However, the scarcity of annotated datasets and linguistic resources hampers progress in this area (*Elaraby & Abdul-Mageed, 2018*). Additionally, most of the research and datasets concerning fake Arabic news are centered on online news articles rather than social media posts, highlighting a significant gap in the study of this problem on platforms like Twitter (*Touahri & Mazroui, 2024*).

In general, datasets for fake news detection can be categorized into two main types: textual datasets and multimodal datasets. Textual datasets primarily consist of articles accompanied by factual labels indicating their news verification status, without any contextual information about the surrounding network environment. In contrast, multimodal datasets include additional information beyond the news content, such as social, visual, and user-related data, offering insights into the social context surrounding the news. Broadly speaking, there is a scarcity of available datasets for fake news detection, with the majority focusing on political news and being confined to the English language (*Capuano et al., 2023*).

### Textual English datasets

Existing datasets in the domain of fake news detection in English can be broadly classified into two main categories based on their content and features: textual datasets and social network datasets. Textual datasets primarily focus on news articles and textual content, emphasizing the analysis of linguistic and semantic information. These datasets typically consist of labeled samples of news articles, where each sample is annotated as either genuine or fake. The LIAR dataset (*Wang, 2017*) and the Reddit Comments dataset (*Setty & Rekve, 2020*) are prominent examples of such textual datasets. These datasets enable researchers to analyze the linguistic and semantic features of fake news and develop text-

| Table 1 "Liar" English article dataset for fake news. | | | |
|---|---|---|---|
| **Name** | **Liar** | **Subject** | **Diverse** |
| **Type** | **Articles** | **Fact-Check** | **Politifact** |
| **Balanced** | **True** | **Total size** | **12,836** |
| Labels | Pants-fire | 1,050 | |
| | False | 2,528 | |
| | Barely-true | 2,108 | |
| | Half-true | 2,638 | |
| | Mostly-true | 2,466 | |
| | True | 2,046 | |

based classification models. Tables 1 and 2 outline the properties of the Liar and Reddit Comments datasets, respectively.

## Social English datasets

On the contrary, social network datasets encompass not only textual content but also incorporate user and network features. These datasets offer a comprehensive perspective on the intricate dynamics of information dissemination and interactions within social media platforms. In addition to textual content, social network datasets provide supplementary contextual information, such as user profiles, follower networks, and engagement metrics. The inclusion of these features allows researchers to explore the influence of network structure, user behavior, and user credibility on the propagation and detection of fake news. Prominent examples of social network datasets extensively used for investigating the dissemination and veracity of rumors on social media include FakeNewsNet (*Shu et al., 2020*) and MediaEval (*Boididou et al., 2018*) datasets. These datasets have significantly contributed to the advancement of fake news detection techniques and have served as benchmarks for assessing the performance of various models. Tables 3 and 4 present the properties and characteristics of the FakeNewsNet and MediaEval datasets, respectively.

## Textual Arabic datasets

The availability of Arabic datasets in the textual (articles) format for fake news detection has been relatively limited compared to the English language resources. However, efforts have been made to develop Arabic textual datasets specifically tailored for veracity analysis. One notable example is called the AFND dataset (*Khalil et al., 2022*), which consists of news articles collected from various Arabic news sources. Table 5 outlines the properties of the AFND dataset. Another significant dataset is AraFacts (*Ali et al., 2021*) the first comprehensive Arabic dataset of naturally occurring claims collected from multiple Arabic fact-checking websites, including Fatabyyano and Misbar. AraFacts consists of 6,223 claims spanning from 2016 to 2020, with each claim accompanied by factual labels and additional metadata such as fact-checking article content, topical categories, and links to posts or web pages associated with the claim (Table 6).

**Table 2 "Reddit Comments" English textual dataset for fake news.**

| Name | Reddit comments | Subject | Diverse |
|---|---|---|---|
| Type | Twitter comments | Fact-check | Snopes, Politifact, Emergent |
| Balanced | False | Total size | 12,597 |
| Labels | Fake | 7,936 | |
| | True | 4,661 | |

**Table 3 "FakeNewsNet" social network English dataset for fake news.**

| Name | FakeNewsNet | Subject | Diverse |
|---|---|---|---|
| Type | Social networks | Fact-check | Politifact, GossipCop |
| Balanced | False | Total size | 23,196 |
| Labels | Fake | 5,755 | |
| | True | 17,441 | |

**Table 4 "MediEval" social network English dataset for fake news.**

| Name | MediaEval | Subject | Diverse |
|---|---|---|---|
| Type | Social networks | Fact-check | Manual |
| Balanced | False | Total size | 15,629 |
| Labels | Fake | 9,404 | |
| | True | 6,225 | |

**Table 5 "AFND" articles Arabic dataset for fake news.**

| Name | AFND | Subject | General news |
|---|---|---|---|
| Type | Articles | Fact-check | Misbar |
| Balanced | False | Total size | 374,543 |
| Labels | Not credible | 167,232 | |
| | Credible | 207,310 | |
| | Undecided | 232,369 | |

**Table 6 "Arafacts" articles Arabic dataset for fake news.**

| Name | AraFacts | Subject | General news |
|---|---|---|---|
| Type | Articles | Fact-check | Misbar, Fatabyyano, FactualAFP, Verify-sy, Maharat-news |
| Balanced | False | Total size | 6,223 |
| Labels | False | 4,037 | |
| | Partly-false | 1,891 | |
| | Unverifiable | 7 | |
| | Sarcasm | 90 | |
| | True | 198 | |

Existing datasets for fake news detection, while valuable, often exhibit notable limitations. Many datasets are either imbalanced, with disproportionate numbers of fake and real news samples, or limited in scope, focusing predominantly on specific content types such as news articles rather than social media posts. Additionally, datasets in Arabic are relatively scarce and frequently lack the comprehensive coverage needed for effective detection across diverse social media platforms. These gaps underscore the pressing need for a robust, balanced dataset tailored specifically for Arabic social media. The VERA-ARAB dataset addresses these limitations by offering a well-balanced and extensive resource designed to enhance fake news detection within the Arabic-speaking social media context. Its comprehensive nature and focus on social media posts not only bridge existing gaps but also provide a valuable tool for improving the accuracy and applicability of machine learning models in this domain.

## DATA ACQUISITION METHODOLOGY

This section provides a comprehensive explanation of the methodology adopted for the construction of the VERA-ARAB dataset. Figure 1 shows the overall methodology of constructed VERA-ARAB. First, the rigorous approach employed for news verification by professional fact-checkers, which involves carefully assessing the credibility of the claims included in the dataset. Additionally, the steps undertaken for claim expansion to retrieve relative tweets and annotate them as either fake or true are explained. Furthermore, the process of gathering the tweets information from Twitter platform is elaborated upon. By providing a detailed account of these methodological steps, transparency and reproducibility in the construction of the VERA-ARAB database are ensured, contributing to its reliability and usability for future work to increase the size of the dataset.

### Fact-checking

For the fact-checking process, interaction was established with Misbar (https://www.misbar.com/twitter), a platform dedicated to verifying Arab news and tweets using evidence-based methods. Misbar employs a rigorous fact-checking methodology, which involves a thorough investigation of claims by analyzing credible sources, consulting experts, and conducting in-depth research. Over the course of approximately 1 year, from August 8, 2022, to September 3, 2023, a total of 884 claims were extracted from Misbar's fact-checking platform. These claims encompass a wide range of domains including sports, politics, armed conflicts, public security, and others that provide a valuable *corpus* for our research on fake news detection in the Arabic language over social media.

### Annotation process

The annotation process involved a systematic approach to curating a balanced dataset of fake and real news tweets. To begin for each claim, relevant search terms and criteria were extracted to identify both fake and real news within a specific time range. These search terms were carefully chosen to capture a diverse range of fake and factual content. These search terms were then utilized to collect a preliminary set of tweets that potentially contained fake news or real news. To ensure the accuracy and reliability of the dataset, a

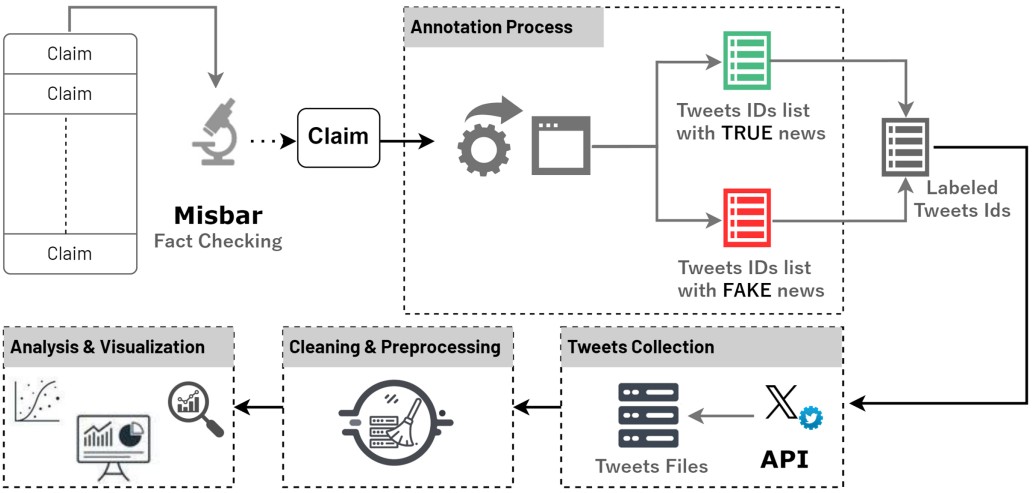

**Figure 1** Methodology for extracting claims and retrieving related tweets.

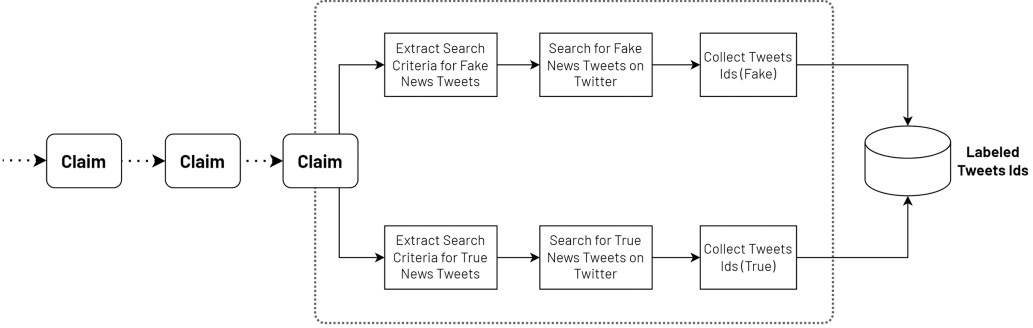

**Figure 2** Annotation process of the VERA-ARAB dataset.

manual annotation process was conducted. The initial collection was expanded by gathering additional tweets that matched the identified fake claims, as well as expanding the set of real news tweets. This approach allowed us to create a comprehensive dataset that represented a balanced distribution of fake and real news.

Throughout the annotation process, each tweet was meticulously reviewed by a team of trained annotators. The annotators carefully assessed the content of each tweet, considering factors such as the accuracy of the information presented, and the presence of any misleading or deceptive elements. Through this rigorous annotation process as shown in Fig. 2, a dataset was successfully compiled, consisting of 11,076 tweets containing fake news and 9,008 tweets containing true news, ensuring a balanced representation of both categories.

## Data collection

The acquisition of actual tweets for our study was accomplished through the utilization of the X API protocol (*X, 2024*). This API provides essential tools and functionalities for accessing and extracting tweets directly from the Twitter platform. By leveraging the X

**Table 7 Tweet raw data fields description.**

| Field | Description |
|---|---|
| tweet_id | The unique identifier of the tweet. |
| text | UTF-8 encoding string of the user's post. |
| possibly_sensitive | Determines whether the content of a tweet may be viewed as sensitive. |
| retweet_count | Describes how many times this tweet has been retweeted. |
| reply_count | Indicates the number of responses to this tweet. |
| like_count | Specifies the number of users who have expressed like of this tweet. |
| quote_count | The count of times this tweet has been re-posted by other users, along with their new message or comment. |
| bookmark_count | Specifies the count of user accounts that have chosen to save or bookmark this particular tweet. |
| impression_count | Reflects the number of times the Tweet has been viewed, regardless of whether it is viewed by the same or different users. |
| annotations_type | A list of text values describe the category of annotations extracted from the tweet. |
| annotations_text | A list of corresponding values of annotations types in the tweet. |
| hashtags | A list of hashtags. A hashtag is a word or phrase preceded by the "#" symbol. It is used to organize tweets around specific topics. |
| mentions | A list of directly addressed other users in a tweet. |
| urls | A list of urls attached in the tweet. |
| created_at | A date-time format that represents the time of the tweet creation. |
| edits_remaining | Displays the number of edit opportunities left for this tweet. Users are permitted to edit their tweets for up to 30 min after initial posting, with a maximum of five edits allowed. |
| is_edit_eligible | Indicates whether a tweet is eligible for editing or not. |
| reply_settings | Specifies who can reply to this tweet ("everyone", "mentioned_users", or "followers"). |

API, a diverse array of valuable data was retrieved, including the textual content of tweets, multimedia attachments (such as images and video URLs), as well as user information and associated engagement metrics. The API's flexibility allowed us to specify precise criteria and parameters to filter tweets relevant to our research objectives. The API was configured to extract tweets based on specific tweet IDs, effectively targeting the data collection process. The robust scalability of the X API facilitated the efficient gathering of a substantial volume of tweets for comprehensive analysis.

In addition to the textual content, the API provided access to extensive user information and metrics, including the number of followers, account creation date, and engagement metrics such as retweets and likes. These user-centric metrics allowed us to gain a deeper understanding of each tweet's reach, impact, and potential for dissemination. By incorporating these diverse data points, an enriched dataset was compiled that not only included the content of the tweets but also detailed the characteristics and behaviors of the users who generated them. This comprehensive approach enabled a more nuanced analysis of fake news on social media, considering both the content and its context within the user network.

## Data records and fields

Utilizing the X API described in the previous subsection, annotated tweets were extracted by their IDs. The retrieval API endpoint from the X platform allows for fetching data in batches, with each request returning a maximum of 100 tweets. The retrieved data is

| Table 8 | User raw data fields description. |
|---------|-----------------------------------|
| **Field** | **Description** |
| user_id | The unique identifier of the user who created this tweet. |
| created_at | A date-time format represents the time the user account (who posted the tweet) was created. |
| user_description | Textual content provided by a user in their profile. |
| location | The location information provided by a user in their account profile. While this field may not always represent an actual, precise geographical location, it can still be leveraged for approximate location-based searches or evaluations. |
| protected | Specifies if a user has opted to make tweets private, restricting access only to users they have approved as followers. |
| followers_count | Represents the total count of other users who have chosen to subscribe and receive updates from that user's tweets in their own timeline. |
| following_count | Refers to the number of accounts that this user is following on X. |
| tweet_count | The total number of tweets that have been posted by this user on their account. |
| listed_count | The count of public Twitter lists that other users have added this account to, as Twitter's list feature enables users to create and categorize accounts into customized groups based on shared topics or interests. |
| like_count | The total count of tweets that this user has marked as likes over the lifetime of their account. |
| verified | Indicates if this user has a verified account or not. |

typically provided in the form of a JSON file, which adopts a dictionary structure comprising various key-value pairs. This dictionary includes distinct sections such as "data", "includes" (encompassing "media", "users", and "tweets"), and "errors".

All retrieved tweet data were stored in files and used to construct the VERA-ARAB dataset, mapping each tweet to its corresponding label by its ID. Table 7 lists all tweet-related fields retrieved from the X platform, while Table 8 details the user-related fields, providing descriptions for each field. By systematically mapping each tweet to its corresponding label, the integrity and accuracy of the VERA-ARAB dataset were ensured. This structured approach enabled a thorough and detailed analysis, facilitating deeper insights into the characteristics and dissemination patterns of fake news on social media.

## DATASET ANALYSIS

VERA-ARAB dataset was carefully curated and validated to ensure high quality and reliability for use in research on Arabic social media and misinformation detection. Several key steps were taken in the data collection, preprocessing, and validation processes. Tweets were collected using the Twitter API, targeting tweets related to the prepared claims. The dataset covers a diverse range of subject areas to ensure broad applicability.

### Dataset statistics

Table 9 offers comprehensive statistics and summarizes the key metrics related to the dataset. It provide an overview of the collected claims, corresponding tweets, and users. It encompasses the total number of claims and their associated tweets and users.

To visualize the distribution of annotated tweets, Fig. 3A presents an overall distribution, highlighting the relative frequencies of real and fake news tweets. Figure 3B further breaks down the distribution by illustrating the annotated tweet distribution for each claim individually. These visual representations offer insights into the prevalence and distribution patterns of real and fake news within the dataset.

**Table 9  Overall statistics of VERA-ARB dataset.**

| | | | |
|---|---|---|---|
| Number of claims | 884 | Max. true tweets in one claim | 327 |
| Number of tweets | 20,084 | Max. fake tweets in one claim | 376 |
| True tweets | 9,008 | Unique users count | 13,421 |
| Fake tweets | 11,076 | Tweets of verified users | 965 |
| Min. tweets in one claim | 1 | Min. tweets per user | 1 |
| Max. tweets in one claim | 703 | Max. tweets per user | 131 |
| Possibly sensitive tweets | 776 | | |

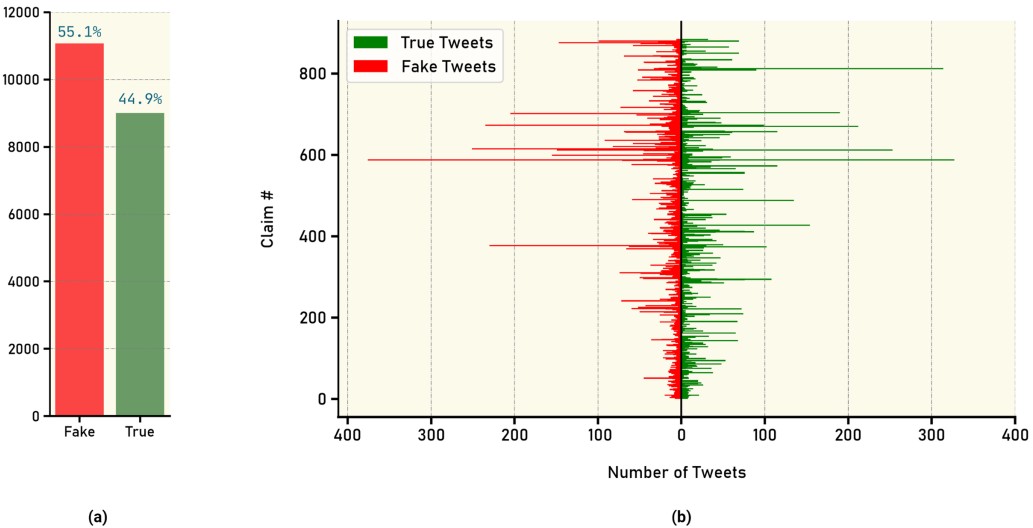

(a)                                                                                     (b)

**Figure 3  Count of tweets labels: (A) overall tweet counts for true *vs.* fake news, (B) number of true and fake news tweets for each claim.**               

Additionally, Fig. 4 provides a graphical representation of the number of tweets posted by each user. Complementing this analysis, Table 10 categorizes the users based on the number of tweets they have posted in the dataset, providing further granularity in the characterization of user behavior. The table shows the analysis of user participation in the VERA-ARAB dataset reveals intriguing insights into the frequency of tweets posted by individual users. The data, represented by rows "Tweets" and "Users" provides a breakdown of the number of users who have participated in specific tweet ranges. For instance, the entry '>1' indicates that 2,653 users have published more than one tweet, while '>2' signifies that 1,192 users have contributed more than two tweets.

Examining the distribution pattern further, the number of users gradually decreases as the tweet count increases. This trend is evident as the tweet count thresholds increase to '>4' (430 users), '>6' (226 users), and '>8' (133 users), respectively. As the tweet count becomes more substantial, the number of users gradually decreases, indicating a smaller subset of highly active participants. For instance, '>10' captures 85 users, '>15' includes 37 users, and '>20' narrows down to 22 users. This analysis provides valuable insights into
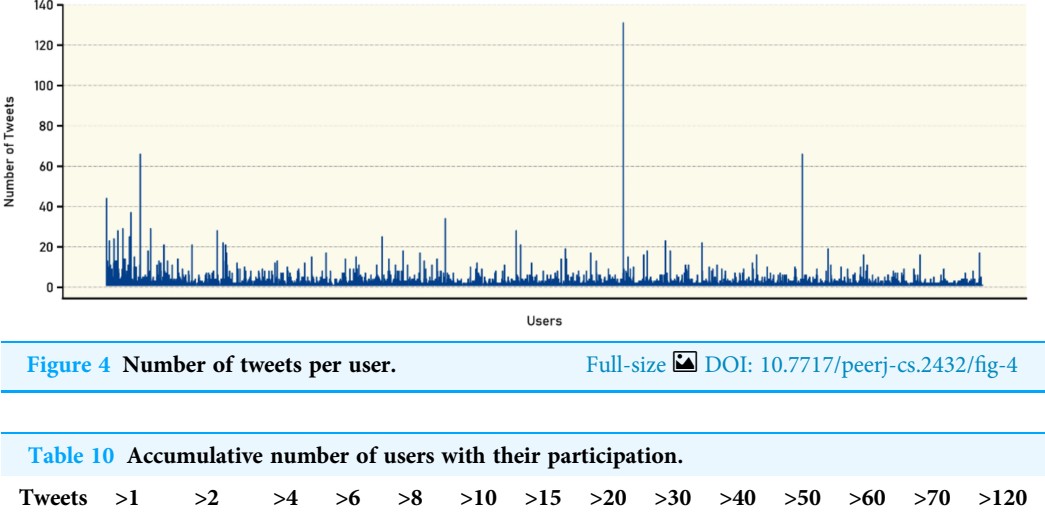

**Figure 4  Number of tweets per user.**

**Table 10  Accumulative number of users with their participation.**

| Tweets | >1 | >2 | >4 | >6 | >8 | >10 | >15 | >20 | >30 | >40 | >50 | >60 | >70 | >120 |
|---|---|---|---|---|---|---|---|---|---|---|---|---|---|---|
| Users | 2,653 | 1,192 | 430 | 226 | 133 | 85 | 37 | 22 | 6 | 4 | 3 | 3 | 1 | 1 |

user engagement and behavior within our dataset. The majority of users exhibit lower levels of participation, contributing only a few tweets. However, a smaller subset of users demonstrates high levels of activity, consistently posting a significant number of tweets. Understanding this distribution of user engagement can provide further context for subsequent analyses, such as identifying influential users or detecting patterns in the dissemination of true and fake news within the dataset.

## Spatiotemporal analysis

Spatiotemporal analysis refers to the examination and interpretation of data that is both spatially and temporally referenced. It involves analyzing patterns, trends, and relationships in data that vary over space and time. This type of analysis combines elements of traditional spatial analysis, which focuses on the geographic distribution of data, with the temporal dimension, which considers the evolution of data over time. In the context of the VERA-ARAB dataset, temporal patterns and geographic distributions of claims, tweets, and user locations were examined.

To perform a comprehensive spatiotemporal analysis, the available data on the timeline of the collected claims from August 8, 2022, to June 18, 2023 were utilized. Additionally, data on the timeline of tweets were collected, showing the distribution count over the years. Chart in Fig. 5 illustrates that the majority of tweets in the years 2022 and 2023 fall within the same timeline, which is expected due to the selective claims being made. Additionally, the chart reveals the existence of tweets predating the year 2022, where claims have been evaluated for their falsification. Through analysis of Twitter platform, previous tweets with similar evaluations or those confirming the validity of the claims were discovered and subsequently included in the dataset. Consequently, we have included them in the dataset.

The heat map in Fig. 6 reveals the number of users whose locations were tracked based on the data recorded in their profiles. It becomes evident that the majority of users are

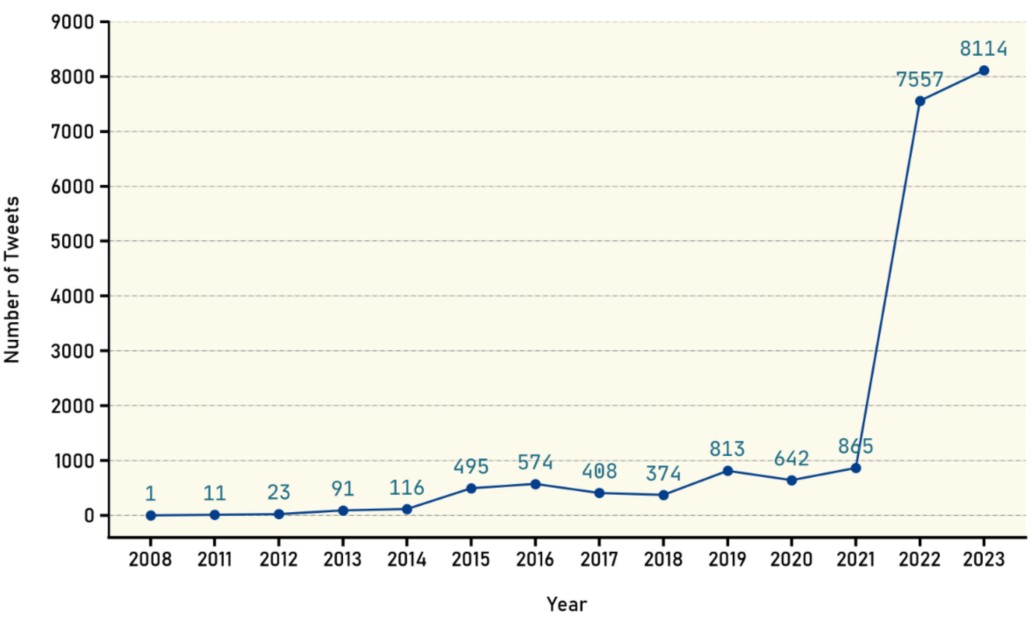

**Figure 5** Distribution of tweets over the years.

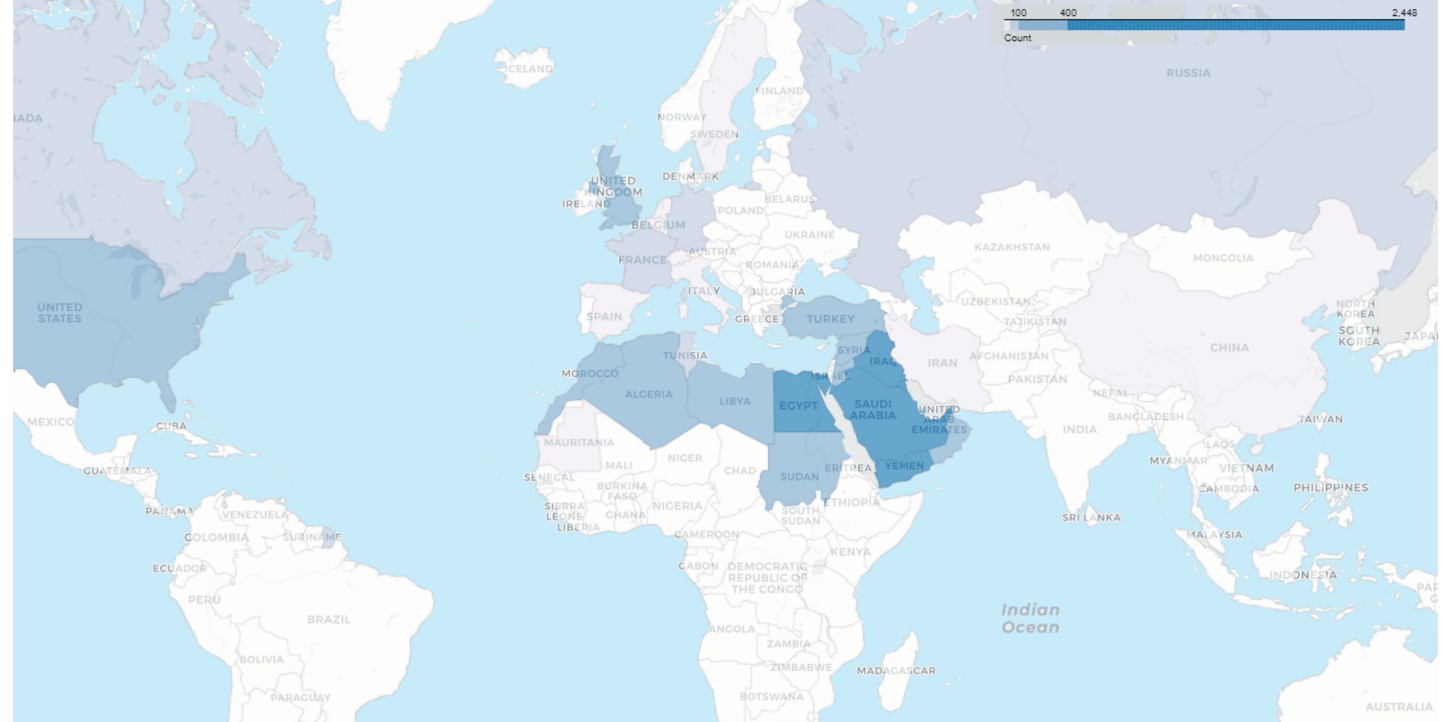

**Figure 6** Geographical distribution of users.

**Table 11 Statistics of users public metrics.**

| | | | |
|---|---|---|---|
| Avg. number of followers | 399,304 | Number of distinct verified users | 164 |
| Avg. number of following | 3,904 | Number of fake tweets by verified users | 103 |
| Avg. number of tweets | 74,049 | Number of true tweets by verified users | 862 |
| Avg. number of listed | 365 | | |

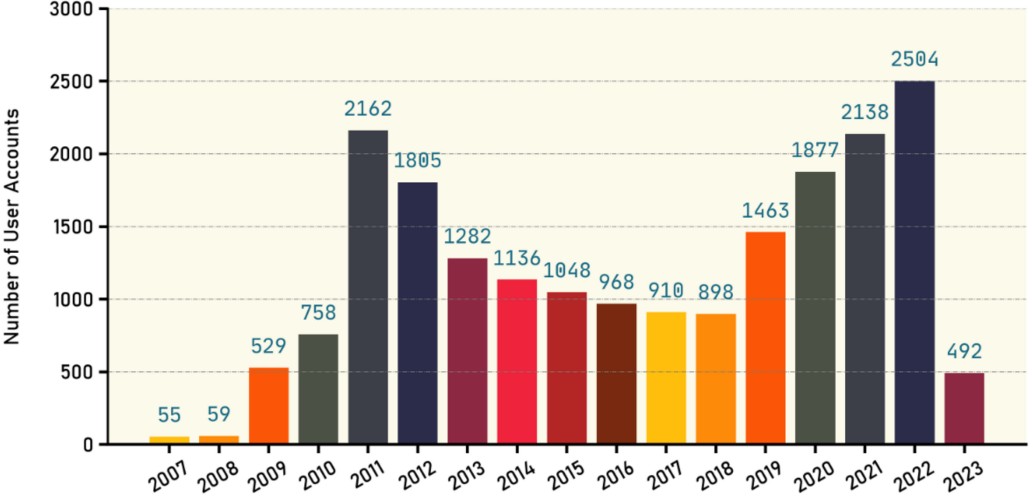

**Figure 7  Number of new user accounts created each year.**

from Arab countries, particularly Saudi Arabia, Egypt, Kuwait, Yemen, and the United Arab Emirates. This outcome is expected due to the dataset being specific to the Arabic language. Following the Arab countries, the United States and the United Kingdom come in the next positions.

## User profile analysis

Analyzing user profiles is of paramount importance in the context of the fake news detection problem. User profiles provide valuable insights into the credibility, reliability, and potential biases of individuals sharing information on social media platforms. Examining user profiles uncovers important indicators that help distinguish between user accounts and those associated with spreading misinformation.

The X platform provides public metrics about users such as followers, following, listed, and tweet count for each profile. Table 11 shows the statistics of such metrics for all user profiles existed in our dataset as well as the number of verified users along with the number of fake tweets and true tweets. Figure 7 gives an overview about the user accounts creation dates. Our database contains 13,421 users, with the majority of these accounts being created since 2009, as illustrated in the figure. It is also evident that there is variation in the number of accounts created during those years, indicating a significant diversity in the ages of those accounts.

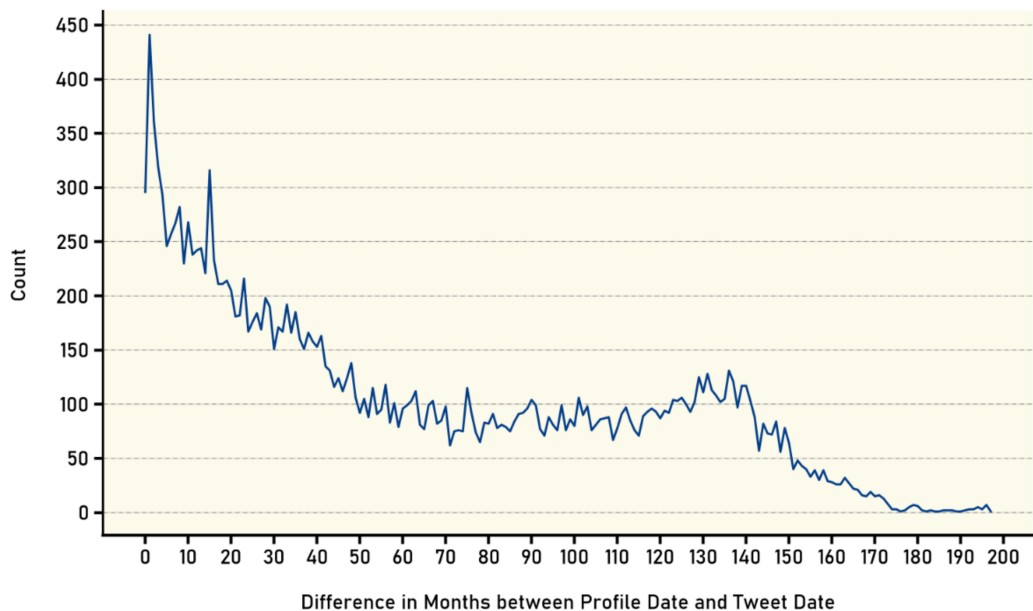

**Figure 8 Age distribution of user accounts at tweet publication.**

Furthermore, the time difference in months between the account creation date and the tweet posting date was calculated for all available tweets. As shown in Fig. 8, the number of tweets was counted based on the age of the account at the time of posting. The results indicate an inverse relationship between the age of the account and the number of posted tweets. The graph reveals that newer accounts tend to have a higher number of tweets at the time of posting with the number of tweets decreasing as the account ages. This inverse relationship can be attributed to several factors, one possible explanation being that newer users on the platform are often more enthusiastic, motivated, and eager to actively participate in the social network community. As they join the platform, the novelty and excitement drive them to share thoughts, engage with others, and contribute to conversations more frequently through tweets.

Additionally, new users often have a smaller network of followers initially, which can result in a higher frequency of tweeting as they strive to establish their presence and gain visibility. They may actively seek interactions, engage with other users, and share content to attract attention and expand their follower base. This increased level of activity and eagerness to connect with others can result in a higher number of tweets during the early stages of their interactive journey. As time progresses and users become more familiar with the platform, they may settle into a regular tweeting pattern or find a balance between tweeting and consuming content. The initial excitement and motivation may gradually diminish, leading to a decrease in the frequency of tweets. Users may have other commitments or priorities that limit the time and energy they can allocate to tweeting, which can contribute to the decline in the number of posted tweets over time. It is worth noting that this observed trend is not universal and may vary among different users. Factors such as personal preferences, interests, and individual tweeting habits can

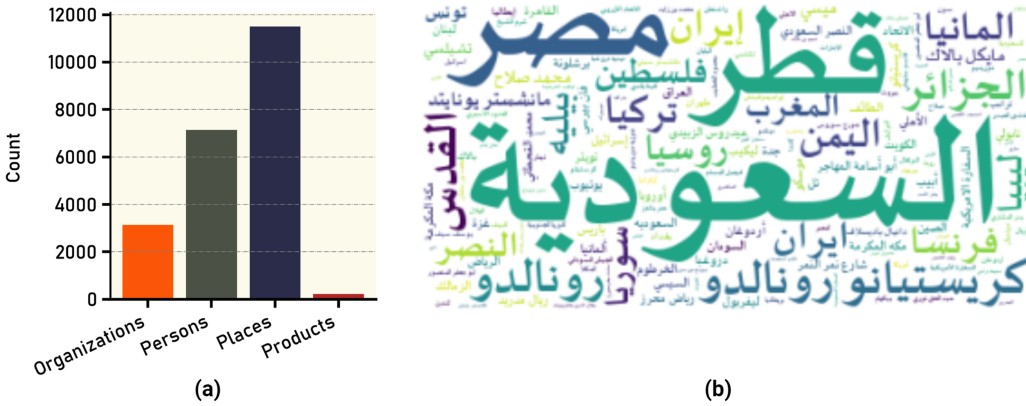

**Figure 9** Text annotations in tweets: (A) frequency of different annotation types; (B) most common annotated words.                           

influence the pattern of tweet frequency and account age. Nonetheless, the general trend suggests that newer accounts tend to exhibit higher tweet activity, while older accounts may experience a gradual reduction in the number of tweets posted.

## Content analysis

Textual content plays a crucial role in extracting meaningful insights from datasets, such as the VERA-ARAB dataset, which specifically focuses on Arabic news content. When extracting tweet data from X platform, the Application Programming Interface (API) provides additional information about the content, such as annotation type and annotation text. The annotation type refers to the classification or categorization assigned to the tweet content based on specific criteria. The classification of the annotation types are organizations, people, places, and products. Figure 9A shows the count of words that annotated in each category. The annotations extracted by X platform produced 3,137 organization, 7,144 person, 11,499 place, and 221 products in the entire dataset. Figure 9B shows the word cloud figure for most frequent annotated words in the entire dataset.

By inspecting all claims and tweets in VERA-ARAB dataset, all tweets were manually annotated and classified into seven domains; religion, natural disaster, public security, armed conflict, public news, politics, and sports news as shown with their proportion in Fig. 10. A significant portion of news domain falls within sports and politics while news related to natural disasters or religions constitutes a smaller proportion in the VERA-ARAB dataset. Additionally, the balance of fake/true label annotations was investigated within each news domain. As shown in Fig. 11, all news domains are nearly balanced with respect to fake/true labels.

### Sports news

Sports news is considered a significant genre of news both in the Arab region and globally. The sports sector itself has become a thriving industry with a distinct economy, particularly in recent decades. As seen in Fig. 10, this domain occupies the largest proportion of the dataset, accounting for approximately 25% with a total of 4,962 tweets (2,586 of which are fake news tweets). This can largely be attributed to the significant

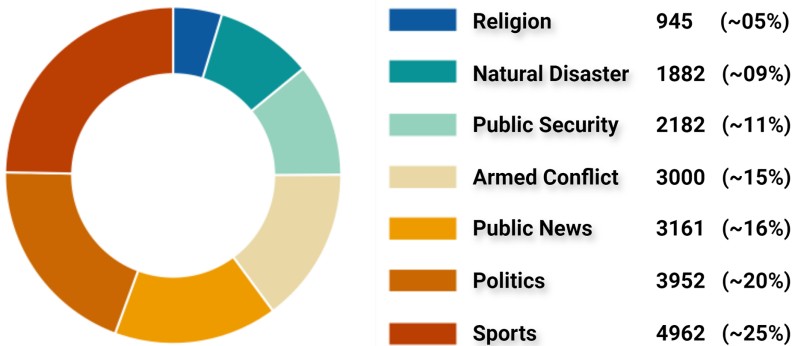

**Figure 10 Classification of tweets by news domain.**

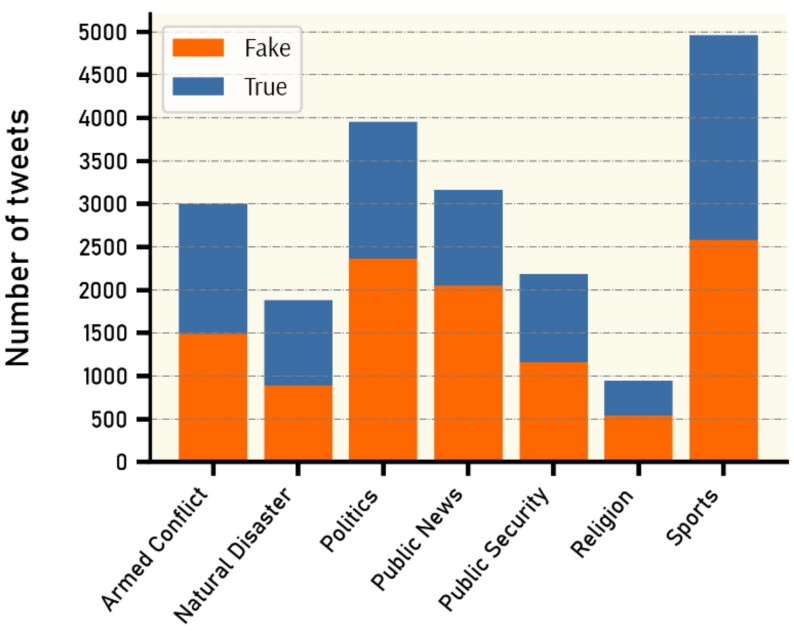

**Figure 11 Fake *vs*. true news distribution by domain.**

attention sports news garners from audiences in the Arab region. Moreover, the data collection period coincided with a notable surge in sporting events in the Arab sports arena. During this period, the first Football World Cup in the Middle East was held in Qatar in 2022. Following the World Cup, there were numerous news reports, including the high-profile transfer of the world-class football player Cristiano Ronaldo to Al-Nassr Club in the Kingdom of Saudi Arabia, among other sports-related news. Figure 12A shows the word cloud of the frequent words in sports news tweets.

### Politics news

It is normal for political news to have a significant presence, dominating daily headlines not only in the Arab region but also worldwide. Approximately 20% of the available news in our dataset consists of political news (2,366 of which are labeled as containing fake news) as shown in Fig. 10. The political events during that period included the Arab

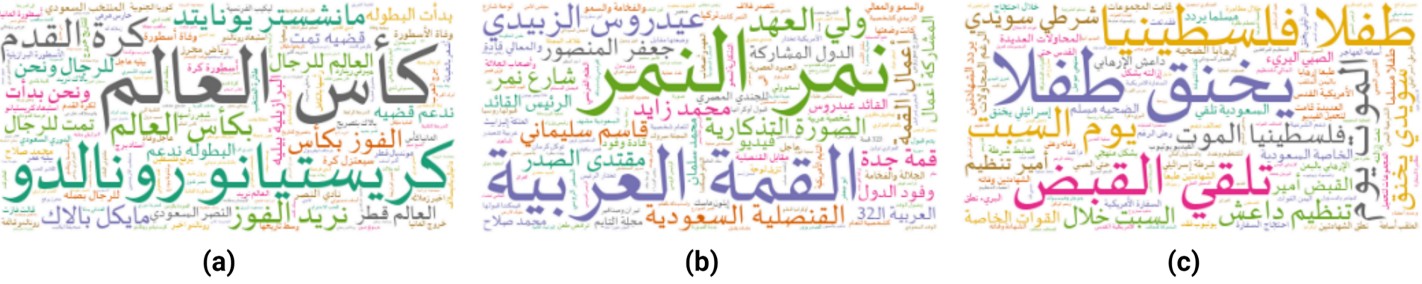

**Figure 12 Words cloud of selected domains: (A) frequent terms in sports news, (B) frequent terms in political news, (C) frequent terms in armed conflict news.**

Summit of the Arab League held in Algeria 2022. Additionally, there were other news reports concerning Iran, Iraq, Yemen, and various political events. Figure 12B shows the words cloud of the frequent words in political news tweets.

### Armed conflict news

The Middle East region is witnessing armed conflicts in various locations. Military conflicts between factions within the Sudanese army, as well as the presence of the armed organization "ISIS", sectarian and military conflicts in Syria, and the Houthi rebels and internal conflict in Yemen, are among the prevalent conflicts. All these conflicts result in the dissemination of various news, including both fabricated and accurate information, through social media platforms. A total number of 3,000 tweets of this domain were collected, of which 1,487 tweets contained fake news, while 1,513 tweets contained accurate news. Figure 12C shows the words cloud of the frequent words in armed conflict news tweets.

### Other domain news

Public news, public security, natural disaster, and religion are other domains exist in our dataset. These types of news encompass a variety of topics, such as celebrity news and general daily life, news pertaining to public security, including daily incidents involving law enforcement agencies. In addition, news concerning natural disasters such as hurricanes, and finally, religious news, such as reports on celebrities converting to Islam and other related matters.

## Dataset limitations and biases

Despite the rigorous construction and comprehensive statistical analysis underpinning the VERA-ARAB dataset, several limitations and potential biases must be acknowledged. One significant concern is the geographical distribution of tweets within the dataset, which may not fully capture the diversity of regions across Arabic-speaking countries. As depicted in Fig. 6, the dataset shows a notable concentration of users from Saudi Arabia and Egypt. This regional imbalance could lead to potential biases, both geographically and dialectally, affecting the generalizability of the findings across different areas and linguistic contexts. Additionally, the dataset exhibits uneven representation of various Arabic dialects, with

some dialects being overrepresented while others are underrepresented. This discrepancy introduces dialectal bias, which may influence the dataset's effectiveness in accurately detecting fake news across the full range of Arabic dialects. The underrepresentation of certain dialects could limit the dataset's ability to generalize findings to all Arabic-speaking populations. These factors underscore the necessity for cautious interpretation of the results and suggest that future research should focus on addressing these biases. Efforts to include a more balanced geographical and dialectal representation would enhance the dataset's inclusivity and improve its utility for comprehensive fake news detection across diverse Arabic-speaking communities.

## FAKE ARABIC NEWS DETECTION

Detecting fake news remains a challenge for researchers, particularly in the context of the Arabic language in social media. Recent studies on fake news detection in Arabic have primarily relied on textual databases consisting of news articles, lacking the characteristics of social networks. For instance, *Fouad, Sabbeh & Medhat (2022)* studied different machine and deep learning techniques on different three Arabic article datasets with textual features only. *Himdi et al. (2022)* conducted a model using their own Arabic article dataset. *Nassif et al. (2022)* used a translated dataset from English to Arabic as well as they tried to collect Arabic news articles from different sources. The absence of a dedicated social media database labeled by experts from multiple domains hinders researchers in this field (Arabic language) from accessing comprehensive resources. Consequently, there is a need for a robust dataset that encompasses social media networks, allowing for a more nuanced analysis incorporating expertise from various domains.

### Preliminaries

This subsection provides a detailed overview of the approach. The problem statement is formally defined, and the key objectives of fake news classification model are outlined. The architectural design and core technical components of the proposed model are then described, with emphasis on the rationale behind the modeling choices. Finally, the evaluation metrics used to assess the model's performance are discussed.

#### Problem statement

The challenge of identifying the presence of fake news on social media platforms has been formulated as a binary classification problem. This conceptual approach mandates the mathematical modeling of the salient entities and their relational structures *via* the employment of a formally defined notational system. The fundamental objects and their corresponding mathematical notation can be summarized as follows (*Shu et al., 2017*): Assume that

$$U \leftarrow \text{set of users } \{u_1, u_2, \ldots u_V\} \tag{1}$$

where $V$ is the number of users

$$P \leftarrow \text{set of posts } \{p_1, p_2, \ldots p_M\} \tag{2}$$

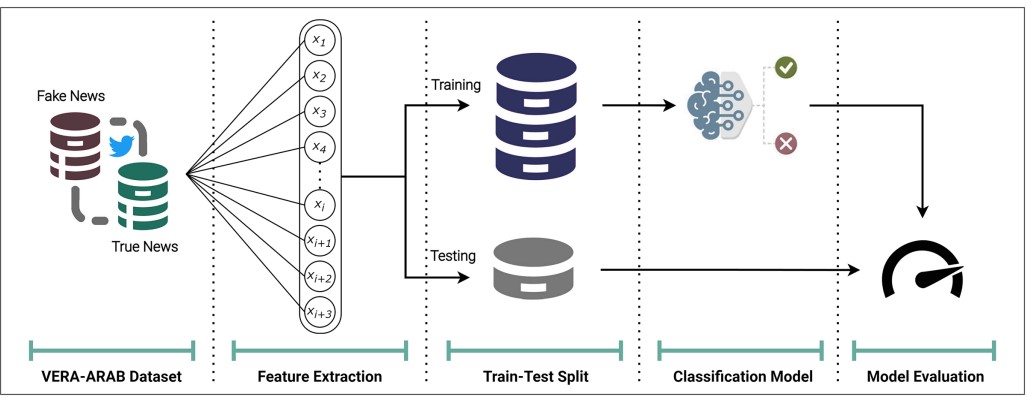

**Figure 13 Overview of the Arabic fake news detection model.**

$M$ is the number of posts

Let $(p_j)$ be a post on a social network that contains specific news published by a specific user $(u_i)$ at time $t$, called engagement $(e)$.

$$T \leftarrow \text{Time period}[1, T] \tag{3}$$
$$E \leftarrow \text{sequence of user engagements} \{e_1, e_2, \ldots e_Z\} \tag{4}$$

where

$$e = (u_i, p_j, t) \tag{5}$$

Given an engagement $(e)$ as an Eq. (5), the fake news detection function $F$ can be formulated as:

$$F(e) = \begin{cases} 1 & : \text{if } p_j \text{ is a piece of fake news,} \\ 0 & : \text{otherwise} \end{cases} \tag{6}$$

This formulation of fake news detection as a binary classification problem is grounded in the theoretical foundations of media bias, as established in prior research (*Gentzkow, Shapiro & Stone, 2015*).

### Model overview

For the fake news classification task, the VERA-ARAB dataset, which contains a balanced distribution of true and fake news labels is utilized to evaluate model performance on an equitable basis. The process begins with the extraction of appropriate comprehensive features as shown in Fig. 13. The dataset is then split into training and testing partitions, with 70% of the samples allocated for model development and the remaining 30% reserved for final evaluation. This partitioning strategy ensures that classifiers are assessed on unseen data, providing a robust estimate of their generalization capabilities. As a baseline, several established machine learning algorithms, including logistic regression and support vector machines, are applied.

To assess the performance of each classifier, a range of appropriate evaluation metrics is employed. Given the class balance in the dataset, accuracy, precision, recall, and F1-score are calculated to provide a nuanced view of the models' ability to correctly identify both true and fake news instances. These complementary performance measures allow for a thorough examination of the strengths and weaknesses of the candidate models. A detailed explanation of each component presented in the following subsections.

## Features extraction

A fundamental aspect of the machine learning pipeline involves transforming raw data into a set of attributes that can be effectively utilized by learning algorithms. Feature extraction significantly impacts the performance of learning algorithms by ensuring the quality and relevance of the input data. High-quality features capture the essential patterns and characteristics of the raw data, enabling the algorithm to learn more effectively and make accurate predictions. Conversely, poor feature extraction may result in the inclusion of irrelevant or noisy data, which can confuse the model and degrade its performance. Additionally, feature extraction often involves reducing the dimensionality of the data, helping to managing computational complexity and prevents overfitting. The features of $d$ dimensions with the following general matrix:

$$X = \begin{bmatrix} x_1^{(1)} & x_2^{(1)} & \cdots & x_d^{(1)} \\ x_1^{(2)} & x_2^{(2)} & \cdots & x_d^{(2)} \\ \vdots & \vdots & \cdots & \vdots \\ x_1^{(M)} & x_2^{(M)} & \cdots & x_d^{(M)} \end{bmatrix}, d: \text{features}, \ M: \text{instances} \tag{7}$$

### Social features extraction

In order to conduct a thorough classification model, it is essential to first extract a comprehensive set of informative features from the raw dataset. This process of feature engineering is a critical step in transforming the unstructured social media data into a format suitable for subsequent modeling and hypothesis testing. The process begins with the extraction of a series of preliminary features directly observed in the dataset, such as fields listed previously in the "Data Records and Fields" section. Figure 14 shows the correlation matrix of those basic features.

Additional features are then derived through the application of domain knowledge and computational techniques, generating secondary attributes that may capture more nuanced aspects of user behavior and content characteristics. These latent features can be extracted from the content, such as the number of words in a tweet, number of charecters, number of hashtags, and other textual attributes. Furthermore, features can be derived from user information such as the age of the user account and the length user name. Figure 15 shows the correlation matrix of those latent features.

Prior to utilizing the extracted feature set for downstream analyses, a series of data preprocessing steps is applied to ensure the features are on a comparable scale and normalized appropriately. First, each numeric feature is standardized by subtracting the

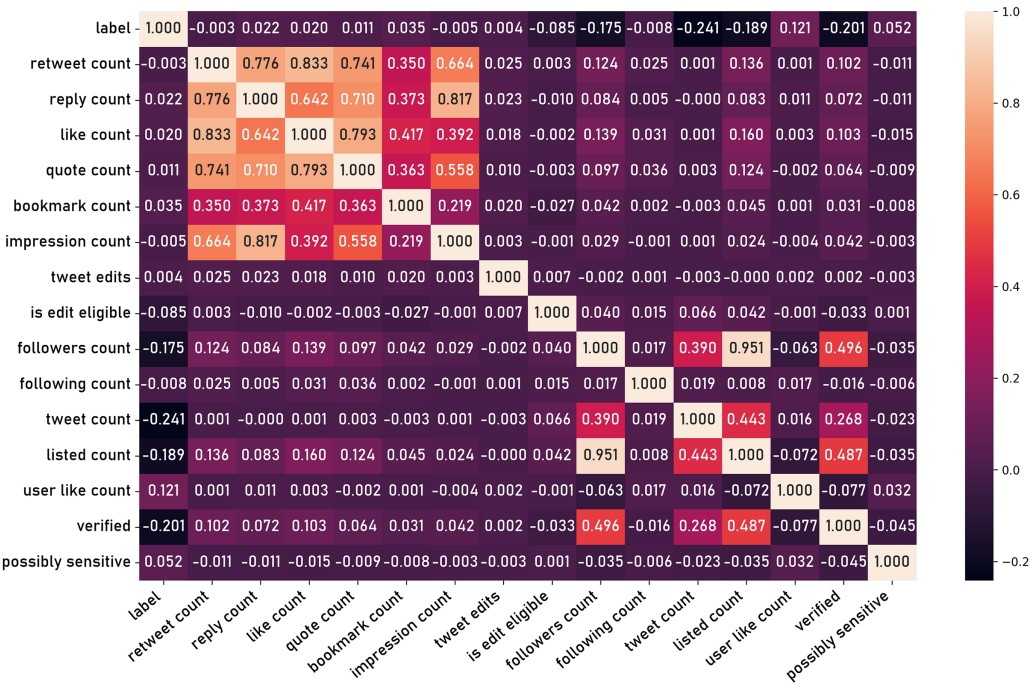

**Figure 14  Correlation matrix of basic features.**     

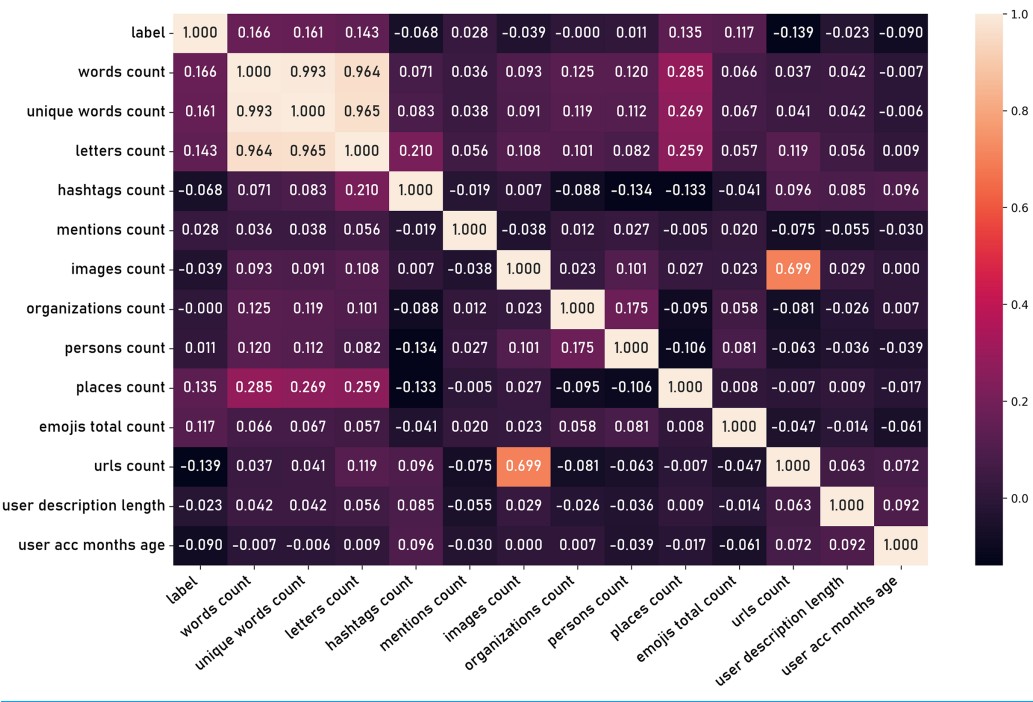

**Figure 15  Correlation matrix of latent features.**     

mean and dividing by the standard deviation. This z-score transformation centers the feature distribution at zero with a unit variance, mitigating the influence of features with larger numerical ranges. Additionally, for features representing inherently bounded quantities, such as sentiment scores or network centrality measures, min-max normalization is applied. This approach linearly scales the feature values to the range of [0, 1], preserving the relative ordering of observations while restricting the values within a common numeric domain. These standardization and normalization techniques are crucial for enabling fair comparisons across the diverse feature set and enhancing the numerical stability of any subsequent modeling procedures.

### Textual features representation

Textual data is a type of unstructured data, and thus, textual features representation is a critical process that transforms unstructured text into structured, quantifiable formats suitable for algorithmic processing. This conversion is pivotal, as it enables models to leverage textual information to discern patterns and make predictions. Common techniques include Bag-of-Words (BoW), Term Frequency-Inverse Document Frequency (TF-IDF), and word embeddings such as Word2Vec. These methods capture various aspects of the text, ranging from simple word occurrence to contextual semantic relationships. In the context of text vectorization for the tweets dataset, let $C$ represent the set of extracted text from the posts. $C = \{c_1, c_2, \cdots, c_M\}$, where $c_i = \text{text}(p_i)$ is the text part of the tweet, and $W = \{w_1, w_2, \cdots, w_K\}$ be the vocabulary of all unique words in the whole dataset.

**Bag of Words (BoW)** simplifies text by representing it as an unordered collection of words, disregarding grammar, syntax, and word order. In this model, each document is converted into a vector, where the length of the vector corresponds to the size of the vocabulary across the entire *corpus*. Each position in the vector reflects the frequency or presence of a specific word within the document. This approach captures the significance of individual terms while remaining computationally efficient, making it a popular choice for text classification tasks. For each post text $c_i$, the BoW model creates a vector $X^{(i)} = [x_1^{(i)}, x_2^{(i)}, \cdots, x_M^{(i)}]$, where $x_j^{(i)}$ denotes the frequency of the word $w_j$ in the post. Mathematically, this can be represented as:

$$x_j^{(i)} = \text{count}(w_j, c_i) \tag{8}$$

that is, the number of times the word $w_j$ appears in the post content $c_i$.

**Term Frequency-Inverse Document Frequency (TF-IDF)** is an advanced textual feature representation technique used in natural language processing to evaluate the importance of a word within a document relative to a dataset. TF-IDF combines two statistical measures: term frequency (TF), which quantifies the occurrence of a term within a document (post text), and inverse document frequency (IDF), which assesses the rarity of the term across the entire dataset. In this context, TF is calculated as the number of times a word appears in a post, divided by the total number of words in that post.

$$TF(w, c) = \log[count(w, c) + 1] \tag{9}$$

IDF is computed as the logarithm of the total number of posts divided by the number of posts containing the word:

$$IDF(w) = \log\left(\frac{M}{|\{c \in C | w \in c\}|}\right) \tag{10}$$

where $M$ is the total number of posts.

The TF-IDF score for a term is calculated as the product of its TF and IDF values. This technique effectively highlights terms that are significant within a document while reducing the weight of commonly occurring terms across the dataset.

$$TF\text{-}IDF(w, c) = TF(w, c) * IDF(w) \tag{11}$$

For each post text $c_i$, the TF-IDF model creates a vector:

$$X^{(i)} = \left[TF\text{-}IDF_{(w_1, c_i)}, TF\text{-}IDF_{(w_2, c_i)}, \cdots, TF\text{-}IDF_{(w_M, c_i)}\right] \tag{12}$$

### Word embedding

Embedding textual features vectors, commonly referred to as word embeddings, is a fundamental technique in natural language processing (NLP) that involves mapping words or phrases from a vocabulary into continuous vector spaces. These embeddings capture semantic meanings by positioning semantically similar words closer together in the vector space. Techniques such as Word2Vec, GloVe (Global Vectors for Word Representation), and FastText have revolutionized this field by enabling models to effectively understand context, polysemy, and syntactic nuances (*Mikolov et al., 2013*; *Pennington, Socher & Manning, 2014*; *Bojanowski et al., 2017*). Word2Vec, for instance, employs a neural network to learn word associations from large corpora using either the continuous bag-of-words (CBOW) or skip-gram models, optimizing word representation in context. GloVe, on the other hand, constructs vectors based on statistical information extracted from word co-occurrence matrices, effectively balancing both local and global information.

When it comes to low-resource languages, the challenges intensify due to the lack of extensive corpora required to train robust word embeddings. Traditional methods that rely on large datasets become infeasible, necessitating alternative approaches such as transfer learning, multilingual embeddings, and cross-lingual models. Techniques like multilingual BERT (mBERT) and XLM-R (Cross-lingual Language Model-RoBERTa) have demonstrated promise in addressing these limitations by leveraging shared subword structures across languages, enabling the learning of representations that generalize effectively even with limited data (*Devlin et al., 2018*; *Conneau et al., 2019*).

AraVec is a set of pre-trained Arabic word embedding models specifically designed to enhance natural language processing (NLP) tasks for the Arabic language. Developed by *Soliman, Eissa & El-Beltagy (2017)*, AraVec provides rich semantic representations for Arabic words by leveraging large Arabic text corpora to train word embeddings using Word2Vec. The models are available in various forms, including Continuous Bag of Words (CBOW) and Skip-gram, and cover different text domains such as Wikipedia,

Twitter, and Web pages. AraVec has significantly contributed to Arabic NLP by providing robust and accurate word vectors that capture the linguistic nuances of Arabic, facilitating tasks such as text classification, sentiment analysis, and named entity recognition. This resource is particularly valuable given the complexity and richness of the Arabic language, including its diverse dialects and morphological variations.

## Classifiers algorithms

For experimentation, a selection of well-established machine learning algorithms were utilized as baseline models for comparison. The choice of these algorithms was guided by (*Wu et al., 2008*), which identifies and discusses the ten leading algorithms in the field of data mining.

### Traditional classifiers

**Naive Bayes classifier (NBC)** is a probabilistic machine learning algorithm based on Bayes' Theorem, assuming independence between predictors. The algorithm calculates the posterior probability of each class given a set of features, selecting the class with the highest probability as the prediction. Its simplicity, efficiency, and low computational cost make it a popular choice for initial classification tasks. The naive Bayes classifier performs well with small datasets and is used in various domains, including medical diagnosis and sentiment analysis (*Rish, 2001*). Prediction function for NBC binary classification with $K$ features:

$$\hat{F}(X) = 1 \text{ if } \left[ P(1) * \prod_{i=1}^{K} P(x_i|1) \right] > \left[ P(0) * \prod_{i=1}^{K} P(x_i|0) \right] \tag{13}$$

Loss function:

$$L = -\sum_{i=1}^{M} \left[ \log(F) + \sum_{j=1}^{K} \log(\hat{F}) \right] \tag{14}$$

**Logistic regression (LR)** represents a fundamental supervised machine learning classification algorithm. This technique models the probability of a binary outcome as a function of one or more predictor variables. Logistic regression applies a logistic sigmoid activation to a linear combination of features, constraining the output to the interval [0, 1] which can be interpreted as the estimated probability of the positive class (*Dreiseitl & Ohno-Machado, 2002*).

Prediction function:

$$\hat{F}(X) = 1 \text{ if } \left( \frac{1}{1 + e^{-y}} \right) \geq 0.5, \ y = \beta_0 + \sum_{i=1}^{K} \beta_i * x_i \tag{15}$$

Loss function:

$$L = \frac{-1}{M} \left[ \left( F * \log(\hat{F}) \right) + (1 - F) * \log\left(1 - \hat{F}\right) \right] \tag{16}$$

**Support vector machines (SVM)** is another supervised learning method for binary classification, which aims to find the optimal hyperplane that maximally separates the positive and negative class instances in the feature space. By casting the classification task as an optimization problem to identify the decision boundary with the largest margin, SVMs can effectively model complex nonlinear relationships without relying on explicit feature transformations (*Suthaharan & Suthaharan, 2016*).

Prediction function:

$$\hat{F}(X) = 1 \text{ if } \left[ \beta_0 + \sum_{i=1}^{K} \beta_i * x_i \right] \geq 0 \tag{17}$$

The optimization:

$$\min_{\beta} \frac{1}{2} ||\beta||^2, \text{ subject to } [\mathrm{F} * (\beta \cdot X + \beta_0)] \geq 1 \tag{18}$$

Loss function (Hinge loss):

$$L = \sum_{i=1}^{M} \max \left( 0, 1 - F * (\beta \cdot X + \beta_0) \right) \tag{19}$$

**K-nearest neighbors (KNN)** is a simple, yet powerful, supervised learning algorithm used for classification and regression tasks. It operates on the principle that similar data points can be found in close proximity to each other. Given a new data point, KNN identifies the $k$ training samples closest in distance and assigns the class label based on the majority vote of these nearest neighbors. The algorithm does not make any assumptions about the underlying data distribution, making it a non-parametric method (*Guo et al., 2003*).

Prediction function:

$$\hat{F}(X) = \arg \max_{c \in \{0,1\}} \sum_{i \in N_k(X)} 1 * \left( F^{(i)} == c \right) \tag{20}$$

### Ensemble classifiers

**Adaptive boosting (AdaBoost)** is an ensemble learning algorithm that enhances the performance of weak classifiers by combining them into a robust composite model (*Freund & Schapire, 1997*). AdaBoost works by iteratively adjusting the weights of misclassified instances, thereby focusing more on difficult cases in subsequent rounds.

**Random Forest (RF)** is an ensemble learning method extensively used in classification and regression tasks due to its robustness and effectiveness. It constructs multiple decision trees during training and merges them to improve the predictive performance and control overfitting. Each tree is trained on a random subset of the data, and a majority vote or averaging is used for the final prediction (*Breiman, 2001*).

**Gradient boosting (GBoost)** is an ensemble machine learning technique used for regression and classification tasks. It builds models in a stage-wise fashion by optimizing a loss function. In each stage, the algorithm fits a new model to the residual errors of the previous model, effectively focusing on the data points that were previously mispredicted. This iterative process results in a strong predictive model composed of an ensemble of weaker models, typically decision trees. The technique flexibility handling different types of data and robustness against overfitting (*Natekin & Knoll, 2013*).

**Extreme gradient boosting (XGBoost)** is a highly efficient and scalable implementation of gradient boosting algorithms designed for supervised learning problems. It leverages gradient boosting principles to optimize the performance of predictive models by combining the predictions of multiple weak learners, typically decision trees. XGBoost incorporates several enhancements such as regularization to prevent overfitting, parallel processing to improve computational efficiency, and handling of missing data (*Chen & Guestrin, 2016*).

## RESULTS AND DISCUSSION

### Evaluation metrics

Evaluation metrics are essential in machine learning classification models as they provide quantitative measures to assess the performance and effectiveness of the models. These metrics help in determining how well the model is making predictions and identifying areas for improvement, ensuring that the model is robust and reliable.

The confusion matrix is a table (as shown in Table 12) that evaluates the performance of the binary classification model and contains four metrics: true positives (TP), false positives (FP), true negatives (TN), and false negatives (FN). True positives are instances where the model correctly predicts the positive class, indicating successful identification of positive cases. True negatives are instances where the model correctly predicts the negative class, reflecting the model's ability to accurately identify non-positive cases. False positives occur when the model incorrectly predicts the positive class for a negative instance. False negatives happen when the model incorrectly predicts the negative class for a positive instance.

From the construction of the confusion matrix, we utilize the following metrics. Accuracy measures the proportion of correct predictions (both true positives and true negatives) out of the total number of cases.

$$Accuracy = (TP + TN)/(TP + TN + FP + FN) \tag{21}$$

Recall, or sensitivity, measures the proportion of true positive cases that are correctly identified by the model.

$$Recall = (TP)/(TP + FN) \tag{22}$$

Precision measures the proportion of true positive predictions out of all positive predictions made by the model.

**Table 12 Binary classification confusion matrix.**

|  |  | Actual labels | |
|---|---|---|---|
|  |  | True | False |
| Predicted labels | True | True positive (TP) | False positive (FP) |
|  | False | False negative (FN) | True negative (TN) |

$$\text{Precision} = (TP)/(TP + FP) \tag{23}$$

The F1 score is the harmonic mean of precision and recall, providing a single metric that balances both the precision and recall of the model.

$$\text{F1} = (2 * \text{Precision} * \text{Recall})/(\text{Precision} + \text{Recall}) \tag{24}$$

### Experiment (1): social features

In the first experiment, the tweets from VERA-ARAB dataset were preprocessed to extract explicit social criteria features such as retweet count, like count, quote count (Fig. 14). These features were selected for their potential to provide insights into the social dynamics and engagement levels associated with the tweets.

Several classifiers were employed to evaluate performance on the extracted features, including NBC, LR, SVM, AdaBoost, RF, GBoost, KNN, and XGBoost. The configurations for these classifiers were meticulously selected to optimize their performance. Logistic Regression was configured with a maximum of 200 iterations to ensure convergence. The SVM was set with a linear kernel accommodate the assumption of the linear separability. KNN was configured with five neighbors to balance bias-variance trade-off. For ensemble methods, the number of estimators was set as follows: 50 for AdaBoost, 100 for RF, and 100 for GBoost. These configurations were designed to leverage the strengths of each classifier, from simple probabilistic models to complex ensemble techniques, in order to provide a comprehensive analysis of the dataset's classification performance.

In the second phase of the experiment, latent features derived from the tweets dataset were introduced. These latent features were expected to capture the nuanced linguistic patterns and contexts that explicit social criteria might overlook, such as word count, unique word count, hashtag count, url count, and others (Fig. 15). Finally, both explicit social criteria and latent textual features were combined to create a comprehensive feature set. This combined approach aimed to harness the strengths of both feature types: the explicit features offering direct social interaction metrics and the latent features providing deeper semantic insights. The same set of classifiers was employed to evaluate this combined feature set.

The evaluation metrics of experiment (1) are presented in Table 13. When utilizing only basic social features such as retweet count, like count, and quote count, GBoost demonstrated the highest accuracy (74.74%), followed closely by other ensemble models. GBoost also achieved the highest F1 score (78.33%) and AUC (81.46%), indicating its

**Table 13 Machine learning with social features performance.** The bold entries refer to the highest scores for each metric.

| Features | Algorithm | Accuracy | Recall | Precision | F1 | AUC |
|---|---|---|---|---|---|---|
| Basic | NBC | 63.06% | **96.40%** | 60.43% | 74.29% | 69.47% |
| | LR | 66.05% | **93.26%** | 63.08% | 75.25% | 73.32% |
| | SVM | 64.14% | **95.80%** | 61.26% | 74.73% | 71.82% |
| | KNN | 67.94% | 73.83% | 69.93% | 71.83% | 72.09% |
| | AdaBoost | 73.71% | 81.29% | 73.86% | 77.40% | 79.91% |
| | RF | 74.24% | 80.64% | **74.81%** | 77.61% | **81.49%** |
| | GBoost | **74.74%** | 82.43% | 74.61% | **78.33%** | 81.46% |
| | XGBoost | 73.33% | 78.09% | **74.83%** | 76.43% | 81.15% |
| Latent | NBC | 58.55% | 46.13% | 68.71% | 55.20% | 64.56% |
| | LR | 62.10% | 71.94% | 64.03% | 67.76% | 65.82% |
| | SVM | 62.23% | 70.89% | 64.44% | 67.51% | 65.57% |
| | KNN | 64.45% | 68.05% | 67.84% | 67.94% | 68.74% |
| | AdaBoost | 64.29% | 72.45% | 66.22% | 69.20% | 69.61% |
| | RF | 68.50% | 75.15% | 70.11% | 72.54% | 74.88% |
| | GBoost | 66.21% | 73.35% | 68.09% | 70.62% | 71.69% |
| | XGBoost | 67.64% | 73.05% | 69.87% | 71.42% | 73.37% |
| Both | NBC | 64.85% | **93.74%** | 62.09% | 74.70% | 71.62% |
| | LR | 69.17% | 84.17% | 67.86% | 75.14% | 74.16% |
| | SVM | 67.01% | **93.32%** | 63.82% | 75.80% | 73.86% |
| | KNN | 68.29% | 73.53% | 70.47% | 71.97% | 73.26% |
| | AdaBoost | 75.36% | 81.71% | **75.70%** | 78.59% | 82.03% |
| | RF | 78.05% | 84.17% | **77.94%** | 80.93% | 86.13% |
| | GBoost | 77.23% | 83.06% | **77.45%** | 80.16% | 84.82% |
| | XGBoost | 78.43% | 83.18% | 78.97% | 81.02% | 86.14% |

strong ability to handle feature variability and complex interactions within the data. However, introducing latent features derived from tweet information, altered the performance dynamics. In this setting, RF showed the highest accuracy (68.50%), reflecting its capacity to effectively leverage complex feature interactions. Notably, the inclusion of latent features resultred in a decrease in performance compared to to the basic features, with the RF algorithm, yielding the best overall results, where all evaluation metrics averaged around 70%.

The combination of both explicit basic and latent features aimed to capitalize on the strengths of feature type. This comprehensive feature set provided a more informative representation of the tweets, leading to improved performance across all classifiers. XGBoost emerged as the top performer, with the highest accuracy (78.43%), F1 score (81.02%), and AUC (86.14%), underscoring its efficiency in handling large, complex datasets with diverse feature sets. Gradient boosting and Random Forest also maintained strong performance, with accuracies of (77.23%) and (78.05%) respectively, further validating their robustness in ensemble learning.

**Table 14 Machine learning with textual features performance.** The bold entries refer to the highest scores for each metric.

| Features | Algorithm | Accuracy | Recall | Precision | F1 | AUC |
|---|---|---|---|---|---|---|
| BoW | MNB | 88.41% | 87.51% | **91.37%** | 89.40% | 95.57% |
| | LR | **91.62%** | 93.79% | **91.42%** | **92.59%** | **97.04%** |
| | SVM | **90.90%** | 92.03% | **91.70%** | **91.87%** | 95.88% |
| | KNN | 85.87% | 93.01% | 83.55% | 88.03% | 93.33% |
| | AdaBoost | 73.75% | 91.73% | 70.31% | 79.60% | 77.38% |
| | RF | **91.22%** | **95.27%** | **89.65%** | **92.37%** | **97.42%** |
| | GraBoost | 75.93% | **95.42%** | 71.23% | 81.57% | 84.98% |
| | XGBoost | 85.69% | 93.79% | 82.85% | 87.98% | 93.45% |
| TF-IDF | MNB | 88.88% | 92.48% | 88.18% | 90.28% | 95.76% |
| | LR | 89.92% | 94.11% | 88.56% | 91.25% | **96.44%** |
| | SVM | **92.13%** | 94.14% | **91.95%** | **93.04%** | **97.26%** |
| | KNN | 86.58% | **97.06%** | 82.15% | 88.99% | 92.54% |
| | AdaBoost | 73.82% | 92.00% | 70.29% | 79.69% | 77.80% |
| | RF | **90.85%** | **94.68%** | 89.54% | **92.04%** | **97.20%** |
| | GraBoost | 76.74% | **95.00%** | 72.15% | 82.02% | 85.98% |
| | XGBoost | 85.36% | 92.92% | 82.91% | 87.63% | 93.16% |

Interestingly, the performance boost was most pronounced in simpler models like LR and SVM, where accuracies improved to (69.17%) and (67.01%) respectively. This suggests that combining explicit and latent features effectively enhanced the predictive power of these models by providing a richer, more nuanced understanding of the data.

## Experiment (2): textual features

In the second experiment, the focus was on the textual vectorization of the tweets' content to assess how different textual representations impact the performance of the classifiers. The same algorithms from Experiment 1 were employed, with the exception of replacing the NBC with multinomial naive Bayes (MNB), which is better suited for text data. Two methods of textual vectorization were utilized. The BoW method represents text by the frequency of words in the *corpus* without considering word order, transforming each tweet into a vector of word counts. The second method, TF-IDF, is a statistical measure used to evaluate the importance of a word in a document relative to a *corpus*, mitigating the influence of frequently occurring but less informative words.

The results, as shown in Table 14 indicate that LR achieves the highest overall performance when using BoW, with an accuracy of (91.62%), an F1 score of (92.59%), and an AUC of (97.04%). This suggests that LR effectively balances the trade-off between precision and recall. RF also demonstrates strong performance, with an accuracy of (91.22%) and an F1 score of (92.37%). In contrast, AdaBoost shows the lowest performance among the algorithms, with an accuracy of (73.75%) and an F1 score of (79.60%), highlighting its limited effectiveness in this context. The SVM algorithm

**Table 15 AraVec: Arabic embedding vectors performance.** The bold entries refer to the highest scores for each metric.

| Features | Algorithm | Accuracy | Recall | Precision | F1 | AUC |
|---|---|---|---|---|---|---|
| (*n*-grams) (cbow) (100) | LR | 67.19% | 74.72% | 69.06% | 71.78% | 73.85% |
| | SVM | 67.64% | 75.02% | 69.47% | 72.14% | 74.13% |
| | KNN | 84.11% | 88.91% | 83.66% | 86.20% | 91.41% |
| | RF | 83.25% | 88.22% | 82.88% | 85.47% | 91.90% |
| | AdaBoost | 69.04% | 74.99% | 71.12% | 73.01% | 74.66% |
| | GBoost | 74.13% | 81.74% | 74.44% | 77.92% | 81.83% |
| | XGBoost | 82.80% | 86.05% | 83.62% | 84.82% | 90.84% |
| (*n*-grams) (skip-grams) (100) | LR | 69.82% | 77.01% | 71.25% | 74.02% | 75.50% |
| | SVM | 70.10% | 77.13% | 71.54% | 74.23% | 75.62% |
| | KNN | **85.94%** | **89.21%** | **86.11%** | **87.63%** | **92.88%** |
| | RF | 83.94% | 88.05% | 83.98% | 85.96% | 92.22% |
| | AdaBoost | 69.33% | 76.15% | 71.02% | 73.50% | 75.48% |
| | GBoost | 75.73% | 82.43% | 76.09% | 79.13% | 82.91% |
| | XGBoost | 83.40% | 86.65% | 84.10% | 85.35% | 91.66% |
| (uni-gram) (skip-grams) (100) | LR | 69.52% | 76.42% | 71.13% | 73.68% | 75.13% |
| | SVM | 70.68% | 77.13% | 72.24% | 74.60% | 75.35% |
| | KNN | **85.85%** | **89.92%** | 85.50% | **87.65%** | **92.76%** |
| | RF | 83.40% | 87.60% | 83.48% | 85.49% | 92.14% |
| | AdaBoost | 69.40% | 76.15% | 71.10% | 73.54% | 75.03% |
| | GBoost | 75.44% | 82.81% | 75.56% | 79.02% | 83.37% |
| | XGBoost | 83.11% | 86.50% | 83.78% | 85.12% | 92.02% |
| (*n*-grams) (cbow) (300) | LR | 75.15% | 79.93% | 76.58% | 78.22% | 81.57% |
| | SVM | 75.81% | 80.26% | 77.29% | 78.75% | 81.42% |
| | KNN | **85.89%** | **89.33%** | **85.95%** | **87.61%** | 92.46% |
| | RF | 83.58% | 87.93% | 83.53% | 85.67% | 92.29% |
| | AdaBoost | 69.20% | 75.08% | 71.29% | 73.14% | 75.71% |
| | GBoost | 76.32% | 82.46% | 76.84% | 79.55% | 84.18% |
| | XGBoost | 84.11% | 87.66% | 84.47% | 86.04% | 92.19% |
| (*n*-grams) (skip-grams) (300) | LR | 75.68% | 80.73% | 76.87% | 78.75% | 82.34% |
| | SVM | 76.57% | 81.21% | 77.81% | 79.47% | 82.17% |
| | KNN | **87.63%** | **91.47%** | **87.04%** | **89.20%** | **93.72%** |
| | RF | 84.84% | 88.02% | 85.30% | 86.64% | **92.95%** |
| | AdaBoost | 70.85% | 76.84% | 72.56% | 74.64% | 77.74% |
| | GBoost | 78.02% | 84.18% | 78.14% | 81.05% | 85.90% |
| | XGBoost | 85.72% | **89.47%** | **85.60%** | 87.50% | **93.17%** |
| (uni-gram) (cbow) (300) | LR | 76.32% | 80.43% | 77.89% | 79.14% | 82.63% |
| | SVM | 76.31% | 80.37% | 77.90% | 79.12% | 82.44% |
| | KNN | **85.87%** | **89.77%** | **85.62%** | **87.65%** | 92.58% |
| | RF | 84.06% | 88.20% | 84.05% | 86.07% | 92.51% |
| | AdaBoost | 71.38% | 76.48% | 73.38% | 74.90% | 77.67% |
| | GBoost | 77.77% | 84.24% | 77.79% | 80.89% | 85.51% |
| | XGBoost | 85.07% | 88.17% | 85.55% | 86.84% | **92.76%** |

performs comparably to LR and RF when using TF-IDF, with an accuracy of (92.13%) and an AUC of (97.26%).

## Experiment (3): embedding contextual features

In Experiment (3), AraVec (*Soliman, Eissa & El-Beltagy, 2017*), a pretrained word representation specifically designed for Arabic text, was employed to embed Arabic tweets. AraVec offers various models tailored to different corpora, and for this study, the Twitter *corpus* was selected to maintain consistency with the tweet-based dataset. Multiple configurations were explored to assess their impact on the performance of the classification algorithms. These configurations included comparing n-grams with unigrams to capture different levels of contextual information, as well as contrasting the Continuous Bag of Words (CBOW) approach with the Skip-gram model to evaluate their efficacy in capturing semantic relationships. Additionally, two vector sizes 100 and 300 were tested to examine the trade-off between computational efficiency and the richness of word representations. Notably, the Naive Bayes algorithm was excluded from this experiment due to its incompatibility with negative values present in some embedding vectors. These variations were aimed at comprehensively analyzing the influence of different word embedding configurations on the performance of various classification algorithms.

The results in Table 15 shows that KNN consistently outperformed other algorithms, achieving the highest accuracy of (87.63%), an F1 score of (89.20%), and AUC of (93.72%) when using n-grams with Skip-gram at vector size of 300. This suggests that KNN is particularly effective in handling the dense vector representations generated by this configuration, which likely captures more nuanced semantic relationships in the Arabic language data. In contrast, algorithms like AdaBoost generally underperformed, indicating that decision tree-based ensemble methods may struggle with the complexity or sparsity of the vectorized text data. RF and XGBoost also demonstrated strong performance, especially with larger vector sizes, indicating their robustness in handling high-dimensional data. The significant improvement in metrics such as accuracy and F1 when moving from a vector size 100 to 300 in models like RF and XGBoost highlights the importance of vector size in capturing sufficient semantic detail for effective classification.

## DISCUSSION

In our study, we conducted three distinct experiments to evaluate different feature extraction techniques and their impact on model performance. These experiments highlight the nuanced ways in which various feature sets interact with machine learning algorithms, providing valuable insights into their effectiveness.

Experiment 1 focused on social features, including basic metrics such as retweet count, like count, and quote count. As shown in Table 13 and Fig. 16, GBoost achieved the highest accuracy, F1 score, and AUC when using these basic features. This performance underscores GBoost's capacity to handle feature variability and complex interactions effectively. However, when latent features derived from tweet information were introduced, RF emerged as the top performer. This shift suggests that while RF can leverage complex feature interactions well, the inclusion of latent features did not enhance

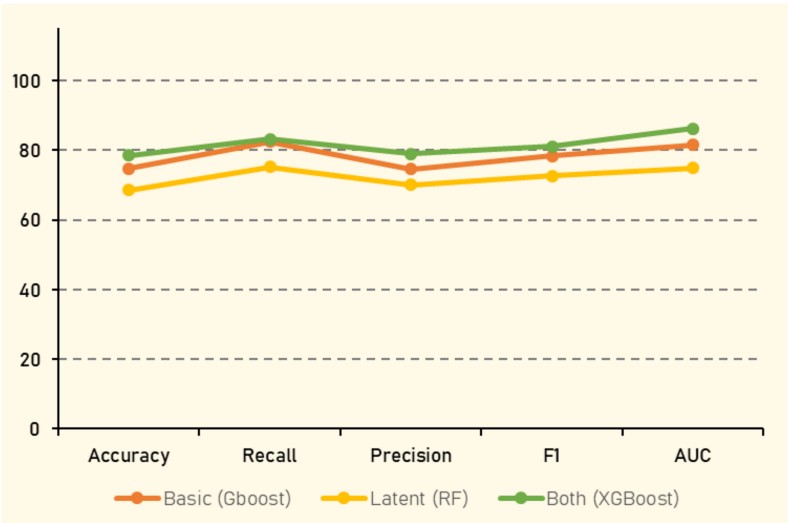

**Figure 16 Best performance results with social features.**

performance as expected, leading to a decrease compared to basic features. The comprehensive feature set combining both basic and latent features aimed to harness the strengths of each. This approach improved performance across all classifiers, with XGBoost showing the highest accuracy (78.43%), F1 score (81.02%), and AUC (86.14%). This improvement indicates that a richer, more nuanced feature representation can enhance predictive power, even for simpler models.

Experiment 2 involved text-based features using BoW and TF-IDF techniques. As detailed in Table 14 and Fig. 17, LR achieved the highest performance with BoW, delivering an accuracy of 91.62%, F1 score of 92.59%, and AUC of 97.04%. This performance reflects LR's effectiveness in balancing precision and recall. Random Forest also performed strongly with BoW, showing an accuracy of 91.22% and F1 score of 92.37%. In contrast, AdaBoost showed the lowest performance with an accuracy of 73.75% and F1 score of 79.60%, indicating its limited effectiveness in this scenario. When using TF-IDF, SVM matched LR and RF in performance, with an accuracy of 92.13% and AUC of 97.26%, highlighting its robustness with this feature representation.

Experiment 3 employed AraVec embeddings with different configurations. As presented in Table 15 and Fig. 18, KNN consistently outperformed other algorithms, achieving the highest accuracy (87.63%), F1 score (89.20%), and AUC (93.72%) with n-grams and a vector size of 300. This suggests that KNN effectively handles the dense vector representations, capturing nuanced semantic relationships in Arabic data. In contrast, AdaBoost generally underperformed, possibly due to challenges in managing the complexity or sparsity of vectorized text data. RF and XGBoost also demonstrated strong performance, particularly with larger vector sizes, indicating their robustness in managing high-dimensional data. In summary, the superior performance of different models with diverse feature engineering techniques can be attributed to their unique strengths in
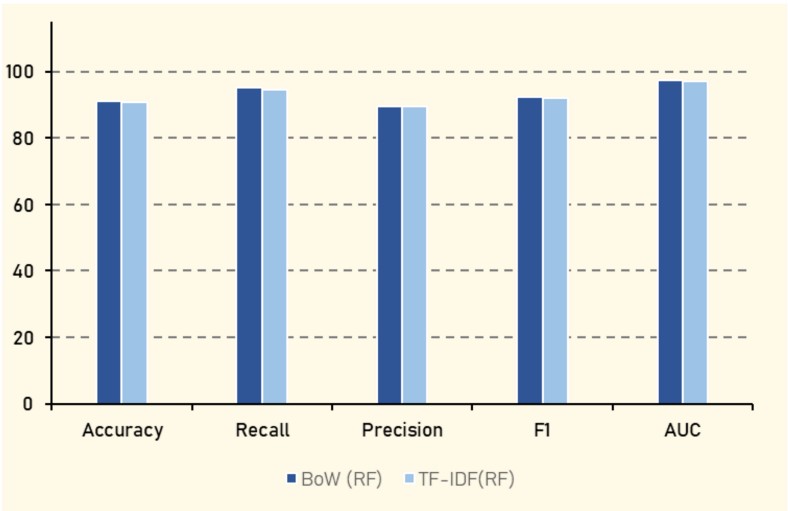

**Figure 17 Best performance results with textual features.**

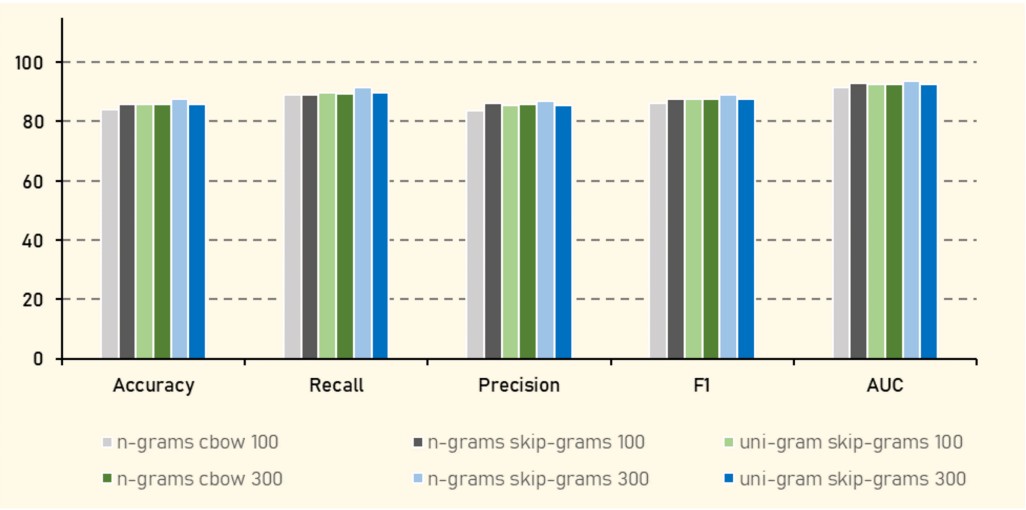

**Figure 18 Best performance results with word embedding features.**

handling various types of data. GBoost excels with simpler features, while RF and XGBoost benefit from more complex feature sets. LR and SVM show adaptability with text-based features, and KNN thrives with dense embeddings. These observations emphasize the need for careful selection of models and features to optimize performance and leverage the strengths of each approach.

## FUTURE WORK AND OTHER APPLICATIONS

For future work, the VERA-ARAB dataset will be expanded using the same annotation methodology, with the goal of establishing this dataset as a solid benchmark for Arabic tweets in the realm of fake news detection. By increasing the size and diversity of the

dataset, more extensive research and development of deep learning models can be enabled, potentially leading to significant improvement in the performance of machine learning classifiers. This expanded dataset will facilitate a deeper understanding of fake news dissemination patterns within Arabic social media and contribute to the development of more robust and accurate detection techniques.

The current state of the VERA-ARAB dataset, which comprises a diverse collection of Arabic news tweets across multiple domains, provides clear classifications distinguishing between tweets containing fake news and those containing true news. Leveraging the rich resources provided by VERA-ARAB, several potential applications can be explored. For instance, the dataset can be utilized for topic modeling to identify prevalent themes and issues within Arabic news tweets. Moreover, the dataset can support research in other dimentions, as outlined below.

## User veracity assessment

In the context of social networks, where information is shared rapidly and extensively, evaluating the credibility and trustworthiness of individual users becomes crucial (*Alrubaian et al., 2021*). Veracity assessment involves examining various indicators, such as a user's social profile (*Al-Qurishi et al., 2018*; *Shu, Wang & Liu, 2018*), posting history, network connections (*Khan & Lee, 2019*), and content characteristics (*Abu-Salih et al., 2019*). Researchers employ diverse methods, and machine learning algorithms, to assess user veracity (*Jia et al., 2019*). The VERA-ARAB dataset, which includes over 13,000 distinct users, provides a valuable resource for conducting experiments and constructing models for user veracity assessment within the context of social networks. The breadth and diversity of the user population contained within the dataset present opportunities for robust analysis and the development of effective veracity assessment frameworks.

## Tweet topic classification

Tweet topic classification involves categorizing content into predefined classes. In a supervised tweet classification task, the process typically begins with a labeled training set, where each tweet is assigned a specific class, such as "Politics" or "Sports". The objective is to develop a classification model capable of accurately assigning a class to new tweet texts (*Daouadi, Rebaï & Amous, 2021*). A key component of such models is the accurate annotation of tweets into relevant classes (*Antypas et al., 2022*). With seven distinct topic classes in the dataset, researchers are provided with the opportunity to construct and develop more robust models in Arabic language.

## Named entity recognition in Arabic tweets

Named entity recognition (NER) is a natural language processing (NLP) task that involves identifying and classifying named entities within text. Named entities refer to specific entities with proper names, such as persons, organizations, locations, dates, and other relevant entities depending on the context (*Li et al., 2020*). Recent studies have focused on addressing this task in the Arabic language, weather local dialects (*Moussa & Mourhir, 2023*) or cross-dialects (*El Elkhbir et al., 2023*). Our dataset contains data about

organizations, people, places, and products in across various Arabic dialects, making it a valuable resource for the named entity recognition task.

## CONCLUSION

In this article, we presented VERA-ARAB, the first large, balanced, multi-domain, and multi-dialectal Arabic tweets dataset containing both fake and true news, verified by the fact-checking experts at Misbar. The dataset comprises approximately 20,000 tweets from 13,000 distinct users, covering 884 different claims. It includes comprehensive information such as news text, user details, and spatiotemporal data, spanning multiple domains including sports and politics.

We provided a detailed data dictionary describing the raw data retrieved from the X platform and conducted a statistical descriptive analysis to reveal insights, explore distribution patterns, and visualize the data according to its type and nature. Additionally, we evaluated the dataset using multiple machine learning classification models with various social and textual features. The results showed promising performance, particularly when using textual features.

In conclusion, we outlined future work in this research area and discussed other potential applications that could leverage the VERA-ARAB dataset, emphasizing its value and versatility for advancing the field of fake news detection in Arabic social media.

### Funding
This work was supported by the Deanship of Scientific Research at King Saud University, Riyadh, Saudi Arabia through the Vice Deanship of Scientific Research Chairs: Chair of Cyber Security. The funders had no role in study design, data collection and analysis, decision to publish, or preparation of the manuscript.

### Grant Disclosures
The following grant information was disclosed by the authors:
Deanship of Scientific Research at King Saud University, Riyadh.
Saudi Arabia through the Vice Deanship of Scientific Research Chairs: Chair of Cyber Security.

### Competing Interests
The authors declare that they have no competing interests.

### Author Contributions
- Mohamed A. Mostafa conceived and designed the experiments, performed the experiments, analyzed the data, performed the computation work, prepared figures and/or tables, gathering data, annotating labels, and approved the final draft.
- Ahmad Almogren conceived and designed the experiments, analyzed the data, authored or reviewed drafts of the article, and approved the final draft.

## Data Availability

The raw data, analysis, and classification are available in the Supplemental Files.

## Supplemental Information

Supplemental information for this article can be found online at http://dx.doi.org/10.7717/peerj-cs.2432#supplemental-information.

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
