# Peer review of "VERA-ARAB: unveiling the Arabic tweets credibility by constructing balanced news dataset for veracity analysis"

_PeerJ Computer Science, doi:10.7717/peerj-cs.2432_

## Round 0.1 · original submission · Minor Revisions

With respect to the reviewers’ comments and my reading of the paper, in my opinion the paper is well-structured and addresses a significant gap in the literature. The dataset and accompanying analysis have the potential to be impactful in the field of computational social science, particularly in Arabic-speaking regions. With minor revisions, it should be suitable for publication. The reviewers have raised several issues concerning the language and clarity. Reviewer 1 has concerns about novelty and clarity of some details, so please address these concerns and provide support for your contribution in your reply. Please respond, point by point, to the issues raised by both the reviewers, making clear any changes in the manuscript.

·

Basic reporting

Check additional comments.

Experimental design

Check additional comments.

Validity of the findings

Check additional comments.

Additional comments

1. Avoid the usage of words like we, our in the paper. 
2. What is VERA-Arab?? authors need to mention it before its first usage.
3. In the introduction line 32, According to Allcott and Gentzkow, and also the same names in reference citation? this mistake is throughout the paper.
4. The term Arab World is unknown or self-made
5. The dataset construction is meticulously verified by an independent fact-checking third party. Which party??
6. Paper structure/organization at the end of the introduction section is missing.
7. Why dataset details and explanations with tables are added in the related work section??
8. The figure 1 is wrong. How is annotation done before extracting tweets using XAPI?
9. Too many data visualization figures were added without showing any relevance to the proposed research problem.
10. Paper is just an application based on hand-crafted and word embedding features and has no novelty.
11. Add graphs to compare results.
12. Justify why different models perform superior to other models using diverse feature engineering techniques.
13. Improve English editing.

Reviewer 2 ·

Basic reporting

Clarity and Language: The paper is well-written, and clear and professional language is used throughout. The flow of information is logical, and key points are well-articulated. However, there are minor grammatical issues and awkward phrasings in some sections that could be improved for better readability.

Context and Literature: The introduction provides a strong context, emphasizing the importance of fake news detection in the Arabic language. The literature review is comprehensive, covering relevant studies in the field of fake news detection, both in English and Arabic. However, there is room for more detailed comparisons with similar datasets and approaches, particularly in terms of their limitations and how VERA-ARAB addresses these gaps.

Figures and Tables: The figures and tables are relevant, well-labeled, and contribute to the overall clarity of the paper. The visualizations of data distributions and model performances are particularly helpful. However, some figures could benefit from more detailed captions to ensure they are fully understandable.

Experimental design

Originality and Scope: The research is original and falls well within the scope of the journal. The authors have identified a gap in the current research landscape—specifically, the lack of robust datasets for fake news detection in Arabic social media—and have addressed it through the development of VERA-ARAB.

Methodology: The methodology is well-documented and thorough, providing sufficient detail for replication. The dataset construction process is meticulously explained, from data collection to the annotation and validation steps. The use of machine learning models to evaluate the dataset is appropriate, though the paper could benefit from a more in-depth discussion of the choice of models and hyperparameters.

Ethical Standards: The paper adheres to high ethical standards, particularly in the careful data verification by independent fact-checkers.

Validity of the findings

Data Robustness: The dataset appears robust and is likely to be a valuable resource for future research. The authors have provided a detailed statistical analysis that supports the validity of their dataset. However, more information on potential biases within the dataset, such as the geographical distribution of tweets or the prevalence of certain dialects, would be beneficial.

Statistical Soundness: The statistical methods used are sound, and the results are presented in a clear and interpretable manner. The authors have effectively demonstrated the potential of the dataset through various machine learning experiments. The performance metrics are appropriately chosen and well-justified.

Conclusions: The conclusions are well-supported by the data and analyses presented. The paper makes a strong case for the significance of VERA-ARAB in advancing fake news detection in the Arabic language.

Additional comments

The paper makes a significant contribution to the field of fake news detection, particularly in the context of Arabic social media. The dataset developed by the authors is a valuable addition to existing resources, and their thorough documentation and analysis set a strong foundation for future research.
Recommendations for Improvement

Language and Clarity: A thorough proofreading of the manuscript is recommended to correct minor grammatical errors and improve the overall clarity of the text.
Comparative Analysis: Including a more detailed comparison with similar datasets would strengthen the paper by highlighting the unique contributions of VERA-ARAB.
Discussion on Biases: A more explicit discussion on potential biases in the dataset (e.g., geographical, dialectal) would add depth to the analysis and provide important context for interpreting the results.

---

## Round 0.2 · accepted · Accept

Reviewer 2 has assessed the revision and is happy that you have addressed all their comments. I have also assessed the revision and am happy with the current version. The article is much improved and meets the high standard required for publication. We appreciate your hard work. Well done.

Reviewer 2 ·

Basic reporting

The revised version addressed the issues raised in the previous review iteration.
It could be accepted.

Experimental design

The revised version addressed the issues raised in the previous review iteration.
It could be accepted.

Validity of the findings

The revised version addressed the issues raised in the previous review iteration.
It could be accepted.

Additional comments

The revised version addressed the issues raised in the previous review iteration.
It could be accepted.